# TopicNet: Semantic Graph-Guided Topic Discovery

**Zhibin Duan, Yishi Xu, Bo Chen**∗**, Dongsheng Wang, Chaojie Wang**
National Laboratory of Radar Signal Processing, Xidian University, Xi'an, China
`xd_zhibin@163.com, bchen@mail.xidian.edu.cn`

**Mingyuan Zhou**
McCombs School of Business, The University of Texas at Austin
`mingyuan.zhou@mccombs.utexas.edu`

## Abstract

Existing deep hierarchical topic models are able to extract semantically meaningful topics from a text corpus in an unsupervised manner and automatically organize them into a topic hierarchy. However, it is unclear how to incorporate prior belief such as knowledge graph to guide the learning of the topic hierarchy. To address this issue, we introduce TopicNet as a deep hierarchical topic model that can inject prior structural knowledge as an inductive bias to influence the learning. TopicNet represents each topic as a Gaussian-distributed embedding vector, projects the topics of all layers into a shared embedding space, and explores both the symmetric and asymmetric similarities between Gaussian embedding vectors to incorporate prior semantic hierarchies. With an auto-encoding variational inference network, the model parameters are optimized by minimizing the evidence lower bound and a regularization term via stochastic gradient descent. Experiments on widely used benchmarks show that TopicNet outperforms related deep topic models on discovering deeper interpretable topics and mining better document representations.

## 1 Introduction

Topic models, which have the ability to uncover the hidden semantic structure in a text corpus, have been widely applied to text analysis. Generally, a topic model is designed to discover a set of semantically-meaningful latent topics from a collection of documents, each of which captures word co-occurrence patterns commonly observed in a document. While vanilla topic models, such as latent Dirichlet allocation (LDA) [1] and Poisson factor analysis (PFA) [2], are able to achieve this goal, a series of their hierarchical extensions [3–12] have been developed in the hope of exploring multi-layer document representations and mining meaningful topic taxonomies. Commonly, these hierarchical topic models aim to learn a hierarchy, in which the latent topics exhibit an increasing level of abstraction when moving towards a deeper hidden layer, as shown in Fig. 1(a). Consequently, it provides users with an intuitive and interpretable way to better understand textual data.

Despite their attractive performance, many existing hierarchical topic models are purely data-driven and incorporate no prior domain knowledge, which may result in some learned topics failing to describe a semantically coherent concept [13], especially for those at a deeper hidden layer [14]. Furthermore, the inflexibility of adding prior knowledge also somewhat limits the applicability of hierarchical topic models, since it is common that a user is interested in a specific topic structure and only focused on information related to it [15]. To address this issue, we assume a structured prior in the form of a predefined topic hierarchy, where each topic is described by a semantic concept, with the concepts at adjacent layers following the hypernym relations, as illustrated in Fig. 1(b). Such a

---

∗Corresponding Author.

35th Conference on Neural Information Processing Systems (NeurIPS 2021).

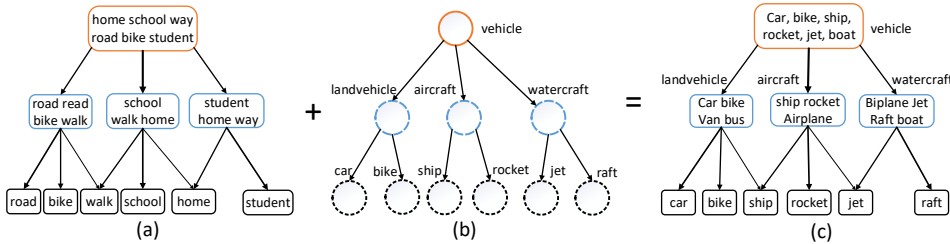

Figure 1: Illustration of (a) topics discovered by a hierarchical topic model, (b) semantic graph constructed by prior knowledge, and (c) topics learned by TopicNet, the proposed knowledge-based hierarchical topic model.

hierarchy can be easily constructed, either by some generic knowledge graph like WordNet [16] or according to the user's customization. However, there are two main challenges: one is how to model this semantic hierarchy, and the other is how to combine it with topic models.

For the first challenge, a general and powerful approach [17] is to learn a distance-preserving mapping, which maps semantically similar topics to nearby points in the embedding space, with a symmetric distance measure (*e.g.*, Euclidean or cosine distance) typically being used. However, the embeddings learned by such a scheme can not perfectly reflect the entailment between concepts. To mitigate this issue, Vendrov et al. [18] exploit the partial order structure of the semantic hierarchy to learn order embeddings that respect this partial order structure. On the other hand, considering that probability distributions are better at capturing uncertainties of concepts than point vectors, Athiwaratkun and Wilson [19] introduce density order embeddings as an improvement to order embeddings. Density order embeddings allows the entailment relationship to be expressed naturally, with general words such as "entity" corresponding to broad distributions that encompass more specific words such as "animal." Through encapsulation of probability densities, it can intuitively reflect the hierarchical structure of semantic concepts, thus offering us an effective way to model the topic hierarchy.

As for the second challenge of combining the constructed topic hierarchy with topic models, there are two major difficulties. For one thing, the structured prior requires modeling the relationship between topics across different layers directly, while hierarchical topic models often assume that the topics at different layers are independently drawn in the prior [6], which makes the fusion of the two a bit far-fetched. For another, since the constructed topic hierarchy can be very deep for large-scale text corpora, an equally deep topic model with multiple latent layers is needed to incorporate with it. However, most existing deep neural topic models are hard to go very deep due to the problem of posterior collapse [14, 20–25]. Fortunately, SawETM [14], a recently proposed neural topic model, not only builds the dependency between topics at different layers in a shared embedding space, but also alleviates the issue of posterior collapse to some extent through a novel network module referred to as Sawtooth Connection (SC), and hence overcoming both difficulties.

In this paper, a novel knowledge-based hierarchical topic model has been proposed, the core idea of which is to represent each topic as a Gaussian embedding, and then project the topics of all layers into a shared embedding space to facilitate the injection of structured prior knowledge into the model. In particular, we first introduce the Gaussian-distributed embeddings to the SawETM, resulting in a variant of SawETM with a stochastic decoder called Gaussian SawETM. To incorporate prior belief into Gaussian SawETM, we then constrain the topics at adjacent layers to encourage them to capture the concepts satisfying the predefined hypernym relations. With auto-encoding variational inference, the entire model is learned in an end-to-end manner by minimizing the evidence lower bound and a regularization term. Extensive experiments demonstrate our model has competitive performance and better interpretability in comparison to most existing hierarchical topic models.

The main contributions of the paper can be summarized as follows:

- To capture semantic uncertainties of words and topics, we propose a novel probabilistic generative model with a stochastic decoder, referred to as Sawtooth Factorial Gaussian Topic Embeddings guided gamma belief network (Gaussian SawETM).

- To incorporate the structured prior knowledge from the real world, we propose TopicNet, a novel knowledge-based hierarchical topic model based on Gaussian SawETM.

- In addition to detailed theoretical analysis, we conduct extensive experiments to verify the effectiveness of the above two models. One of the most appealing properties of TopicNet is

its interpretability. We conduct extensive qualitative evaluation on the quality of the topics discovered by TopicNet, including word embedding, topic embedding, and topic hierarchies.

## 2 Gaussian SawETM

In this section, we elaborate on the construction and inference of Gaussian SawETM, a deep hierarchical topic model that represents words and topics as Gaussian-distributed embeddings [26].

### 2.1 Symmetric similarity: expected likelihood kernel

The distance measure plays a key role in quantifying the similarities between embeddings. A simple choice is taking the inner product between the means of two Gaussians, which, however, does not take the advantage of the semantic uncertainties brought by Gaussian-distributed embeddings. Here we employ the expected likelihood kernel [19, 26, 27] as our similarity measure, which is defined as

$$\mathrm{E}^{(\mathrm{s})}\left(\boldsymbol{\alpha}_i, \boldsymbol{\alpha}_j\right) = \int_{\boldsymbol{x} \in \mathbb{R}^n} \mathcal{N}\left(\boldsymbol{x}; \boldsymbol{\mu}_i, \boldsymbol{\Sigma}_i\right) \mathcal{N}\left(\boldsymbol{x}; \boldsymbol{\mu}_j, \boldsymbol{\Sigma}_j\right) d\boldsymbol{x} = \mathcal{N}\left(\boldsymbol{0}; \boldsymbol{\mu}_i - \boldsymbol{\mu}_j, \boldsymbol{\Sigma}_i + \boldsymbol{\Sigma}_j\right), \quad (1)$$

where $\boldsymbol{\alpha}_i$ is a Gaussian distribution with mean $\boldsymbol{\mu}_i$ and diagonal covariance matrix $\boldsymbol{\Sigma}_i$. As a symmetric similarity function, this kernel considers the impact of semantic uncertainties brought by covariances.

### 2.2 Document decoder with sawtooth factorization and Gaussian embeddings

The generative network (also known as decoder) is one of the core components of topic models. As discussed in Section 1, to build dependency between topics at two adjacent layers and learn a deep topic hierarchy, we draw experience from the Sawtooth Connection (SC) in SawETM. We develop a stochastic decoder by introducing Gaussian-distributed embeddings to better represent the topics in SawETM. Formally, the generative model with $T$ latent layers can be expressed as

$$\boldsymbol{x}_j^{(1)} \sim \mathrm{Pois}(\boldsymbol{\Phi}^{(1)}\boldsymbol{\theta}_j^{(1)}), \left\{\boldsymbol{\theta}_j^{(t)} \sim \mathrm{Gam}(\boldsymbol{\Phi}^{(t+1)}\boldsymbol{\theta}_j^{(t+1)}, 1/c_j^{(t+1)})\right\}_{t=1}^{T-1}, \boldsymbol{\theta}_j^{(T)} \sim \mathrm{Gam}(\boldsymbol{\gamma}, 1/c_j^{(T+1)}),$$

$$\left\{\boldsymbol{\Phi}_k^{(t)} = \mathrm{Softmax}\left(\log\left(\mathrm{E}^{(\mathrm{s})}(\boldsymbol{\alpha}^{(t-1)}, \boldsymbol{\alpha}_k^{(t))})\right)\right)\right\}_{t=1}^T, \left\{\boldsymbol{\alpha}_k^{(t)} \sim \mathcal{N}(\boldsymbol{x}; \boldsymbol{\mu}_k^{(t)}, \boldsymbol{\Sigma}_k^{(t)})\right\}_{t=0}^T, \quad (2)$$

In the above formula, $\boldsymbol{x}_j \in \mathbb{Z}^V$ denotes the word count vector of the $j^{th}$ document, which is factorized as the product of the factor loading matrix $\boldsymbol{\Phi}^{(1)} \in \mathbb{R}_+^{V \times K_1}$ and gamma distributed factor score $\boldsymbol{\theta}_j^{(1)}$ under the Poisson likelihood; and to obtain a multi-layer document representation, the hidden units $\boldsymbol{\theta}_j^{(t)} \in \mathbb{R}_+^{K_t}$ of the $t^{th}$ layer are further factorized into the product of the factor loading $\boldsymbol{\Phi}^{(t+1)} \in \mathbb{R}_+^{K_t \times K_{t+1}}$ and hidden units of the next layer. The top-layer hidden units $\boldsymbol{\theta}_j^{(T)}$ are sampled from a prior distribution. In addition, the $k^{th}$ topic $\boldsymbol{\alpha}_k^{(t)}$ at layer $t$ follows a Gaussian distribution with mean $\boldsymbol{\mu}_k^{(t)}$ and covariance $\boldsymbol{\Sigma}_k^{(t)}$, where the mean vector describes a semantic concept and the covariance matrix reflects its level of abstraction. Note that in the bottom layer $\boldsymbol{\alpha}^{(0)}$ represents the distributed embeddings of words. And $\boldsymbol{\Phi}_k^{(t)}$, used to capture the relationships between topics at two adjacent layers, is calculated based on the topic representations in the shared embedding space, instead of being sampled from a Dirichlet distribution. In particular, in the equation $\mathrm{E}^{(\mathrm{s})}(\cdot)$ refers to the symmetric similarity function defined in Eq. (1).

### 2.3 Document encoder: Weibull upward and downward encoder networks

As presented in Section 2.2, instead of using Gaussian latent variables like most of neural topic models [28], our generative model employs the gamma distributed latent variables that are more suitable for modeling sparse and non-negative document representations. While in sampling based inference, the gamma distribution is commonly used to represent the conditional posterior of these latent variables, the difficulty of reparameterizing a gamma distributed random variable makes it difficult to apply it to an inference network [29, 30]. For this reason, we utilize a Weibull upward-downward variational encoder inspired by the work in Zhang et al. [25, 31]. We let

$$q(\boldsymbol{\theta}_j^{(t)} \,|\, -) = \mathrm{Weibull}(\boldsymbol{k}_j^{(t)}, \boldsymbol{\lambda}_j^{(t)}), \quad (3)$$

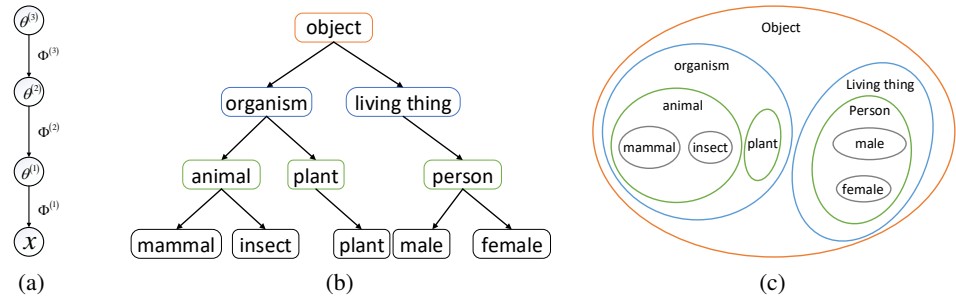

Figure 2: Overviews of (a) generative model of Gaussian SawETM, (b) TopicTree constructed by WordNet, and (c) Density order embeddings where specific entities correspond to concentrated distributions encapsulated in broader distributions of general entities.

where, the parameters $\boldsymbol{k}_j^{(t)}, \boldsymbol{\lambda}_j^{(t)} \in \mathbb{R}^{K_t}$ are deterministically transformed from both the observed document features and the information from the stochastic up-down path $\boldsymbol{\theta}_j^{(t+1)}$. In detail, the inference network can be expressed as

$$
\begin{aligned}
\boldsymbol{h}_j^{'(0)} &= \mathrm{ReLU}(\mathbf{W}_1^{(0)}\boldsymbol{x}_j + \boldsymbol{b}_1^{(0)})) \quad \boldsymbol{h}_j^{'(t)} = \boldsymbol{h}_j^{'(t-1)} + \mathrm{ReLU}(\mathbf{W}_1^{(t)}\boldsymbol{h}_j^{'(t-1)} + \boldsymbol{b}_1^{(t)}), \quad t = 1, \cdots, T, \\
\boldsymbol{h}_j^{(t)} &= \boldsymbol{h}_j^{'(t)} \oplus \boldsymbol{\Phi}^{(t+1)}\boldsymbol{\theta}_j^{(t+1)}, \qquad t = 0, \cdots, T-1, \qquad \boldsymbol{h}_j^{(T)} = \boldsymbol{h}_j^{'(T)}, \\
\boldsymbol{k}_j^{(t)} &= \mathrm{Softplus}(\mathbf{W}_2^{(t)}\boldsymbol{h}_j^{(t)} + \boldsymbol{b}_2^{(t)}), \quad \boldsymbol{\lambda}_j^{(t)} = \mathrm{Softplus}(\mathbf{W}_3^{(t)}\boldsymbol{h}_j^{(t)} + \boldsymbol{b}_3^{(t)}), \quad t = 0, \cdots, T,
\end{aligned}
\tag{4}
$$

where $\{\boldsymbol{b}_i^{(t)}\}_{i=1,t=1}^{3,T} \in \mathbb{R}^{K_t}$, $\{\mathbf{W}_i^{(t)}\}_{i=1,t=1}^{3,T} \in \mathbb{R}^{K_t \times K_{t-1}}$, and $\{\boldsymbol{h}_j^{(t)}\}_{j=1,t=1}^{N,T} \in \mathbb{R}^{K_t}$, $\oplus$ denotes the concatenation in feature dimension, $\mathrm{ReLU}(\cdot) = \max(0, \cdot)$ is the nonlinear activation function, and $\mathrm{Softplus}(\cdot)$ applies $\ln[1 + \exp(\cdot)]$ to each element to ensure positive shape and scale parameters of the Weibull distribution. To reduce the risk of posterior collapse, a skip-connected deterministic upward path [14] is used to obtain the hidden representations $\{\boldsymbol{h}_j^{'(t)}\}_{j=1,t=1}^{N,T}$ of the input $\boldsymbol{x}_j$.

## 2.4 Inference and estimation

Similar to variational auto-encoders (VAEs) [32, 33], the training objective of Gaussian SawETM is to maximize the Evidence Lower Bound (ELBO):

$$
L_{\mathrm{ELBO}} = \sum_{j=1}^{N} \mathbb{E}\left[\ln p(\boldsymbol{x}_j | \boldsymbol{\Phi}^{(1)}, \boldsymbol{\theta}_j^{(1)})\right] - \sum_{j=1}^{N}\sum_{t=1}^{T} \mathbb{E}\left[\ln \frac{q(\boldsymbol{\theta}_j^{(t)}|-)}{p(\boldsymbol{\theta}_j^{(t)}|\boldsymbol{\Phi}^{(t+1)}, \boldsymbol{\theta}_j^{(t+1)})}\right],
\tag{5}
$$

where $q(\boldsymbol{\theta}_j^{(t)}|-)$ is the variational Weibull distribution in Eq. (3), and $p(\boldsymbol{\theta}_j^{(l)})$ is the prior gamma distribution in Eq. (2). The first term is the expected log-likelihood or reconstruction error, while the second term is the Kullback–Leibler (KL) divergence that constrains $q(\boldsymbol{\theta}_j^{(l)})$ to be close to its prior $p(\boldsymbol{\theta}_j^{(l)})$ in the generative model. The analytic KL expression and simple reparameterization of the Weibull distribution make it simple to estimate the gradient of the ELBO, with respect to $\{\boldsymbol{\mu}_k^{(t)}\}_{t=0}^{T}$ and $\{\boldsymbol{\Sigma}_k^{(t)}\}_{t=0}^{T}$ of Gaussian embeddings and other parameters in the inference network.

# 3 TopicNet: Semantic graph-guided topic discovery

In Section 2, a novel hierarchical topic model called Gaussian SawETM is proposed to discover meaningful topics organized into a hierarchy. Below we describe our solution to incorporate prior belief, $e.g.$, knowledge graph, into Gaussian SawETM to guide the learning of the topic hierarchy.

## 3.1 TopicTree: Semantic graph constructed by prior knowledge

Finding an appropriate way to encode prior knowledge is especially important for realizing the incorporation of prior belief into topic models. Due to the hierarchical structure of deep topic models,

correspondingly, we represent prior knowledge in the form of a hierarchical semantic graph and we call it TopicTree. Particularly, in the TopicTree each node is described by a semantic concept and the concepts of nodes at two adjacent layers follow the entailment relations, which results in a bottom-up process of gradual abstraction, as shown in Fig. 2(b). Besides, it is worth mentioning that some generic knowledge graphs such as WordNet [16] provide us a convenient way to build the TopicTree, since the hypernyms or hyponyms of any given word can be easily found in them.

### 3.2 Partial order: Asymmetric entailment relationship

As described in Section 3.1, we organize prior knowledge into a hierarchical semantic graph so that it can be incorporated into our Gaussian SawETM naturally. Specifically, for a constructed TopicTree with depth $T$, we build a Gaussian SawETM with $T$ latent layers corresponding to it, then we match the topics of each layer in the Gaussian SawETM with the semantic concepts of the corresponding layer in the TopicTree, *i.e.*, each topic has a corresponding semantic concept to describe it. Most significantly, the relationships between semantic concepts in the TopicTree should also be injected to guide the learning of the topic hierarchy. Since the Gaussian SawETM projects the topics of all layers to a shared embedding space, such a goal can be achieved by imposing constraints on the topic representations in the shared embedding space.

Inspired by Vendrov et al. [18], the semantic hierarchy (akin to the hypernym relation between words) of the TopicTree can be seen as a special case of the partial order structure. In detail, a partial order over a set of points $X$ is a binary relation $\preceq$ such that for $a, b, c \in X$, the following properties hold: 1) $a \preceq a$ (reflexively); 2) if $a \preceq b$ and $b \preceq a$ then $a = b$ (anti-symmetry); 3) if $a \preceq b$ and $b \preceq c$ then $a \preceq c$ (transitivity). Vendrov et al. [18] propose learning such asymmetric relationships with order embeddings: they represent semantic concepts as vectors of non-negative coordinates, and stipulate that smaller coordinates imply higher position in the partial order and correspond to a more general concept. For example, the following expression defines an ordered pair $(x, y)$ of vectors in $\mathbb{R}_+^N$:

$$\boldsymbol{x} \preceq \boldsymbol{y} \text{ if and only if } \bigwedge_{i=1}^{N} \boldsymbol{x}_i \geq \boldsymbol{y}_i. \tag{6}$$

Since the order embedding condition defined by Eq. (6) is too restrictive to satisfy by imposing a hard constraint, Vendrov et al. [18] relax the condition and turn to seek an approximate order-embedding. In particular, for an ordered pair of points, they apply a penalty that is formulated as:

$$\boldsymbol{E}(\boldsymbol{x}, \boldsymbol{y}) = \| \max(0, \boldsymbol{y} - \boldsymbol{x}) \|^2. \tag{7}$$

Note that this imposes a strong prior on the embeddings, which could encourage the learned relations to satisfy the partial order properties of transitivity and antisymmetry.

Although Gaussian-distributed embeddings are used to represent topics in our Gaussian SawETM, we find that they are also natural at capturing the partial order structure (or the semantic hierarchy of the TopicTree) [19]. For instance, for two topics following the hypernym relation, their Gaussian-distributed embeddings $\boldsymbol{\alpha}_i$ and $\boldsymbol{\alpha}_j$ form an ordered pair. Assuming that $\boldsymbol{\alpha}_i$ represents the specific topic and $\boldsymbol{\alpha}_j$ represents the general one, then we have $\boldsymbol{\alpha}_i \preceq \boldsymbol{\alpha}_j$, meanwhile we expect that $\boldsymbol{\alpha}_j$ corresponds to a broad distribution that encompasses $\boldsymbol{\alpha}_i$. To meet this expectation, the KL divergence between $\boldsymbol{\alpha}_i$ and $\boldsymbol{\alpha}_j$ provides us a ready-made tool:

$$\boldsymbol{E}^{\text{(a)}}(\boldsymbol{\alpha}_i, \boldsymbol{\alpha}_j) = D_{KL}(\boldsymbol{N}_i \parallel \boldsymbol{N}_j) = \int_{\boldsymbol{x} \in \mathbb{R}^n} N(\boldsymbol{x}; \boldsymbol{\mu}_i, \boldsymbol{\Sigma}_i) \log(N(\boldsymbol{x}; \boldsymbol{\mu}_i, \boldsymbol{\Sigma}_i) / N(\boldsymbol{x}; \boldsymbol{\mu}_j, \boldsymbol{\Sigma}_j)) \, d\boldsymbol{x}$$

$$= \frac{1}{2} \left( \text{tr} \left( \boldsymbol{\Sigma}_j^{-1} \boldsymbol{\Sigma}_i \right) + (\boldsymbol{\mu}_i - \boldsymbol{\mu}_j)^{\mathsf{T}} \boldsymbol{\Sigma}_j^{-1} (\boldsymbol{\mu}_i - \boldsymbol{\mu}_j) - d + \log \left( \det(\boldsymbol{\Sigma}_j) / \det(\boldsymbol{\Sigma}_i) \right) \right)$$

However, using $\text{E}^{\text{(a)}}(\boldsymbol{\alpha}_i, \boldsymbol{\alpha}_j)$ directly as a penalty for violating the partial order is undesirable. Since the KL divergence has a property that $\text{E}^{\text{(a)}}(\boldsymbol{\alpha}_i, \boldsymbol{\alpha}_j) = 0$ if and only if $\boldsymbol{\alpha}_i = \boldsymbol{\alpha}_j$, which means the penalty is zero only if $\boldsymbol{\alpha}_i = \boldsymbol{\alpha}_j$, but we expect the penalty should also be zero when $\boldsymbol{\alpha}_i \neq \boldsymbol{\alpha}_j$ and $\boldsymbol{\alpha}_i \preceq \boldsymbol{\alpha}_j$. For this reason, following Athiwaratkun and Wilson [19], we consider using a thresholded divergence as our penalty for order violation:

$$d_\gamma(\boldsymbol{\alpha}_i, \boldsymbol{\alpha}_j) = \max \left( 0, \boldsymbol{E}^{\text{(a)}}(\boldsymbol{\alpha}_i, \boldsymbol{\alpha}_j) - \gamma \right). \tag{8}$$

So this penalty can be zero if $\boldsymbol{\alpha}_i$ is properly encapsulated in $\boldsymbol{\alpha}_j$. Nevertheless, note that we no longer have the strict partial order with such a penalty. In other words, the transitivity and anti-symmetry are not guaranteed. Therefore, actually our learned relations also respect an approximate order structure.

### 3.3 Objective: ELBO with a prior regularity

Essentially, TopicNet is a topic model, so the ELBO still plays a leading role in the training objective. At the same time, a regularization term is also necessary for injecting hierarchical prior knowledge to guide the topic discovery. Therefore, the final objective of TopicNet can be written as:

$$L_{\text{TopicNet}} = L_{\text{ELBO}} + \beta L_{\text{prior}}, \tag{9}$$

where $\beta$ is a hyper-parameter used to balance the impact of these two terms. Particularly, for the ELBO term $L_{\text{ELBO}}$, it is the same as defined by Eq. (5). As for the regularization term $L_{\text{prior}}$, we cannot only constrain topics that follow the hypernym relation between two adjacent layers, but the topics that do not follow the hypernym relation should also be considered. This is consistent with the notion of positive and negative sample pairs in contrastive learning. Based on this idea, we use a max-margin loss to encourage the largest penalty between positive sample pairs to be below the smallest penalty between negative sample pairs, which is defined as:

$$L_j^{(t+1)} = \max(0, m - \max \{d_\gamma(i,j)\}_{i \in D} + \min \{d_\gamma(i,j)\}_{i \notin D}). \tag{10}$$

This represents the penalty for the $j^{th}$ topic at $(t+1)^{th}$ layer, and $D$ is the set of hyponym topics of topic $j$. For $i \in D$, $(i,j)$ forms a positive pair, for $i \notin D$, $(i,j)$ forms a negative pair. Parameter $m$ is the margin that determines the degree to which the positive and negative pairs are separated. Considering the penalties for all topics in each layer, we obtain the prior regularization term.

$$L_{\text{prior}} = \sum_{t=0}^{T-1} \sum_{j=0}^{K^{(t+1)}} L_j^{(t+1)}. \tag{11}$$

With the final objective, TopicNet learns the topics influenced by both the word co-occurrence information and prior semantic knowledge, thus discovering more interpretable topics. Note that, in our experiments, the hyper-parameters are set as $m = 10.0$ and $\beta = 1.0$.

### 3.4 Model Properties

In this part, we recap several good properties of TopicNet: Uncertainty, Interpretability, and Flexibility.

**Effectiveness of Gaussian-distributed embeddings:** Mapping a topic to a distribution instead of a vector brings many advantages, including: on one hand, better capturing uncertainty about a representation, where the mean indicates the semantics of a topic and the covariance matrix reflects its level of abstraction; on the other hand, expressing asymmetric relationships more naturally than point vectors, which is beneficial to modeling the partial order structure of semantic hierarchy [19, 26].

**Hierarchical interpretable topics:** In conventional hierarchical topic models, the semantics of a topic is reflected by its distributed words, and we have to summarize a semantically coherent concept manually by post-hoc visualization. In some cases, the topic is too abstract to get a semantically coherent concept, which makes it hard to interpret. Such cases are common in deeper hidden layers. While in TopicNet, we know in advance what semantic concept a certain topic corresponds to, since it is encoded from prior knowledge and used to guide the topic discovery process of the model as a kind of supervision information. Therefore, once the model is trained, the words distributed under each topic are semantically related to the concept that describes it, as shown in Fig. 6. From this perspective, our TopicNet has better interpretability.

**Flexibility to incorporate prior belief:** Traditional topic models learn topics only by capturing word co-occurrence information from data, which do not consider the possible prior domain knowledge (*e.g.*, Knowledge Graph like WordNet [34]). However, TopicNet provides a flexible framework to incorporate the structured prior knowledge into hierarchical topic models.

## 4 Experiments

We conducted extensive experiments to verify the effectiveness of Gaussian SawETM and TopicNet. Our code is available at `https://github.com/BoChenGroup/TopicNet`.

Table 1: Comparison of per-heldout-word perplexity on three different datasets.

| Method | Depth | 20NG | RCV1 | Wiki | Method | Depth | 20NG | RCV1 | Wiki |
|---|---|---|---|---|---|---|---|---|---|
| LDA [1] | 1 | 735 | 942 | 1553 | DNTM-WHAI [25] | 1 | 762 | 952 | 1657 |
| AVITM [38] | 1 | 784 | 968 | 1703 | DNTM-WHAI [25] | 5 | 726 | 906 | 1595 |
| ETM [35] | 1 | 742 | 951 | 1581 | DNTM-WHAI [25] | 15 | 724 | 902 | 1592 |
| DLDA-Gibbs [36] | 1 | 702 | - | - | DNTM-SawETM [14] | 1 | 718 | 908 | 1536 |
| DLDA-Gibbs [36] | 5 | 678 | - | - | DNTM-SawETM [14] | 5 | 688 | 873 | 1503 |
| DLDA-Gibbs [36] | 15 | 670 | - | - | DNTM-SawETM [14] | 15 | 684 | 864 | 1492 |
| DLDA-TLASGR [7] | 1 | 714 | 912 | 1455 | DNTM-GaussSawETM | 1 | 714 | 904 | 1529 |
| DLDA-TLASGR [7] | 5 | 684 | 877 | 1432 | DNTM-GaussSawETM | 5 | 685 | 866 | 1477 |
| DLDA-TLASGR [7] | 15 | 673 | 842 | 1421 | DNTM-GaussSawETM | 15 | 678 | 857 | 1452 |

**Baseline methods and their settings:** We compare Gaussian SawETM with the state-of-the-art topic models: 1. LDA Group, including: latent Dirichlet allocation(**LDA**) [1], which is a basic probabilistic topic model; LDA with Products of Experts (**AVITM**) [28], which replaces the mixture model in LDA with a product of experts and uses the AVI for training; Embedding Topic Model (**ETM**) [35], a generative model that incorporates word embeddings into traditional topic model. 2. DLDA Group, including: Deep LDA inferred by Gibbs sampling (**DLDA-Gibbs**) [36] and by TLASGR-MCMC (**DLDA-TLASGR**) [7]. 3. Deep Neural Topic Model (DNTM) Group, including Weibull Hybrid Autoencoding Inference model (**WHAI**) [25], which employs a deep variational encoder to infer hierarchical document representations, and Sawtooth Factorial Embedded Topic Model (**SawETM**) [14], where topics are modeled as learnable deterministic vectors. For all baselines, we use their official default parameters with best reported settings.

**Datasets** Our experiments are conducted on four widely-used benchmark datasets, including 20Newsgroups (20NG), Reuters Corpus Volume I (RCV1), Wikipedia (Wiki), and a subset of the Reuters-21578 dataset (R8), varying in scale and document length. 20NG, with a vocabulary of 2,000 words, has 20 classes and was split into 11,314 training and 7,532 test documents. RCV1 consists of 804,414 documents with a vocabulary size of 8,000. Wiki, with a vocabulary size of 10,000, consists of 3 million documents randomly downloaded from Wikipedia using the script provided by Hoffman et al. [37]. R8, with a vocabulary size of 10,000, has 8 classes and was split into 5,485 training and 2,189 test documents. The summary statistics of these datasets and other implementation details (such as dataset preprocessing and length statistics) are described in the Appendix.

## 4.1 Unsupervised learning for document representation

**Per-heldout-word perplexity:** To measure predictive quality of the proposed model, we calculate the per-heldout-word perplexity (PPL) [7], on three regular document datasets, $e.g.$, 20NG, RCV1, and Wiki. For deep topic models in the DLDA and DNTM groups, we report the PPL with 3 different stochastic layers $T \in \{1, 5, 15\}$. Specifically, for a 15-layer model, the topic size from bottom to top is set as $\mathbf{K} = [256, 224, 192, 160, 128, 112, 96, 80, 64, 56, 48, 40, 32, 16, 8]$, and the detailed description can be found in the Appendix. For each corpus, we randomly select 80% of the word tokens from each document to form a training matrix X, holding out the remaining 20% to form a testing matrix Y. We use X to train the model and calculate the per-held-word perplexity as

$$\exp \left\{ -\frac{1}{y_{..}} \sum_{v=1}^{V} \sum_{n=1}^{N} y_{vn} \ln \frac{\sum_{s=1}^{S} \sum_{k=1}^{K^1} \phi_{vk}^{(1)s} \theta_{kn}^{(1)s}}{\sum_{s=1}^{S} \sum_{v=1}^{V} \sum_{k=1}^{K^1} \phi_{vk}^{(1)s} \theta_{kn}^{(1)s}} \right\},$$

where $S$ is the total number of collected samples and $y_{..} = \sum_{v=1}^{V} \sum_{n=1}^{N} y_{vn}$. As shown in Tab. 1, overall, DLDA-Gibbs perform best in terms of PPL, which is not surprising as they can sample from the true posteriors given a sufficiently large number of Gibbs sampling iterations. DLDA-TLASGR is a mini-batch based algorithm that is much more scalable in training than DLDA-Gibbs, at the expense of slightly degraded performance in out-of-sample prediction. Apart from these deep probabilistic topic models, Gaussian SawETM outperforms other neural topic models in all datasets. In particular, compared with Gaussian based single layer neural topic model, WHAI performs better, which can be attributed to two aspects, the first is the effectiveness of Weibull distribution in modeling sparse and non-negative document latent representations, and the second is aggregating the information in both the higher (prior) layer and that upward propagated to the current layer via inference network. Benefiting from the embedding decoder and the Sawtooth Connection that builds the dependency

Table 2: Comparison of document classification and clustering performance.

| Methods | Depth | Classification | | Clustering | | | |
|---|---|---|---|---|---|---|---|
| | | 20NG | R8 | 20NG | | R8 | |
| | | | | ACC | NMI | ACC | NMI |
| LDA [1] | 1 | 72.6±0.3 | 88.4±0.5 | 46.5±0.2 | 45.1±0.4 | 51.4±0.4 | 40.5±0.3 |
| NVITM [38] | 1 | 71.5±0.4 | 87.3±0.2 | 48.3±0.5 | 46.4±0.3 | 52.5±0.2 | 41.2±0.4 |
| ETM [35] | 1 | 71.9±0.2 | 87.9±0.4 | 49.8±0.6 | 48.4±0.5 | 55.3±0.4 | 42.3±0.2 |
| PGBN [6] | 1 | 74.2±0.5 | 90.8±0.5 | 46.6±0.5 | 45.4±0.6 | 51.7±0.4 | 40.7±0.4 |
| PGBN [6] | 4 | 76.0±0.3 | 91.6±0.4 | 48.2±0.2 | 46.5±0.5 | 54.4±0.4 | 41.2±0.3 |
| WHAI [25] | 1 | 72.8±0.3 | 85.3±0.2 | 49.4±0.5 | 46.5±0.4 | 57.9±0.5 | 42.3±0.4 |
| WHAI [25] | 4 | 73.7±0.5 | 88.4±0.3 | 49.5±0.4 | 47.0±0.3 | 60.4±0.4 | 44.1±0.2 |
| SawETM [14] | 1 | 70.5±0.3 | 88.6±0.2 | 50.2±0.2 | 48.7±0.3 | 61.2±0.4 | 43.4±0.4 |
| SawETM [14] | 4 | 71.1±0.2 | 90.5±0.3 | 51.2±0.3 | 50.7±0.2 | 63.8±0.3 | 45.9±0.5 |
| Gaussian SawETM | 1 | 72.8±0.3 | 89.8±0.4 | 51.3±0.4 | 50.6±0.2 | 61.5±0.3 | 44.1±0.2 |
| Gaussian SawETM | 4 | 75.2±0.4 | 91.0±0.3 | 53.0±0.3 | 51.3±0.2 | 65.2±0.4 | 47.0±0.3 |
| TopicNet | 1 | 71.6±0.2 | 88.6±0.3 | 49.5±0.3 | 46.3±0.4 | 61.0±0.2 | 43.2±0.3 |
| TopicNet | 4 | 74.3±0.5 | 89.5±0.4 | 50.8± 0.3 | 47.8±0.4 | 64.2±0.3 | 46.5±0.2 |

of hierarchical topics, SawETM performs better than WHAI, especially at deeper layers. However, modeling topics with point vectors, SawETM cannot capture the uncertainties of topics. Gaussian SawETM represents the topics with Gaussian-distributed embeddings to capture the uncertainties, and achieves promising improvements over SawETM, especially for modeling complex and big corpora such as Wiki. Note that although the methods in DLDA group get better performance, they require iteratively sampling to infer latent document representations in the test stage, which limits their application [25]. By contrast, Gaussian SawETM can infer latent representations via direct projection, which makes it both scalable to large corpora and fast in out-of-sample prediction.

**Document Classification & Clustering**   Document-topic distributions can be viewed as unsupervised document representations, to evaluate their quality, we perform document classification and clustering tasks. In detail, we first use the trained topic models to extract the latent representations of the testing documents and then use logistic regression to predict the label and use k-means to predict the clusters. we measure the classification performance by classification accuracy, and clustering performance by accuracy (ACC) and normalized mutual information (NMI), both of which are the higher the better. we test the model performance on 20NG and R8, where the document labels are considered. Different from the perplexity experiments, a larger vocabulary of 20,000 words is used for 20News to achieve better performance. The network structure is defined by TopicTree, which is introduced in Section 3.1, with the details deferred to the Appendix. The comparison results are summarized in Tab. 2. Benefiting from the use of Gaussian distributions on topics, Gaussian SawETM outperforms the other neural topic models on not only document classification but also clustering, demonstrating its effectiveness of extracting document latent representations. Note that Gaussian SawETM acquires promising improvements compared to SawETM in the 20NG dataset, showing the potential for modeling more complex data. With the predefined knowledge prior, TopicNet does not perform better, possibly because the introduced prior information does not well align with the end task. We leave the refinement of prior information for specific tasks to future study.

## 4.2   Interpretable topic discovery

Apart from excelling at document representation learning, another appealing characteristic of TopicNet is that it can discover interpretable hierarchical topics. In this section, we perform the topic discovery experiment with a 7-layer TopicNet trained on 20NG. The network structure is constructed corresponding to the structure of TopicTree, which is set as [411, 316, 231, 130, 46, 11, 2].

**Prior knowledge and corresponding learned topic:**   We show six example topics and their corresponding children nodes in Tab. 3, which illustrates how a particular concept is being represented in the 20 newsgroups. For example, the "military_action" concept, which has "assault", "defense", and "war" as its children nodes, has become tightly connected to the Waco Siege heavily discussed in this dataset; the "macromolecule" concept, which has "oil" as its single child node, has been represented by a topic related to vehicle, which is interesting as the 20newsgroups dataset has the

Table 3: Semantic graph and corresponding learned topic on the bottom layer.

| Topic | Concept | Children word nodes | Topic words |
|-------|---------|---------------------|-------------|
| 11 | military_action | assault defense war | gun fbi guns koresh batf
waco assault children compound weapons |
| 131 | coding_system | address shareware
software unix windows | windows dos driver microsoft
drivers running applications unix using network |
| 178 | macromolecule | oil | car bike cars engine
oil saturn ford riding miles road |
| 186 | polity | government | government law rights state
police right federal shall crime court laws |
| 296 | atmosphere | sky | space launch nasa gov
satellite earth mission shuttle hst orbit sky |
| 298 | color | black blue
brown color green red | color green blue red
black brown led subject showed lines |

following two newsgropus— "rec.autos' and "rec.motorcycles"—but not a newsgroup clearly related to biology and chemistry; and the "Atmosphere" concept, which has "Sky" as its single child node, has been represented by a topic related to NASA, reflecting the fact that "sci.space" is one of the 20 news groups.

**Topic quality:** Two metrics are considered here to evaluate the topic quality. The first metric is topic coherence, which is computed by taking the average Normalized Pointwise Mutual Information (NPMI) of the top 10 words of each topic [39]. It provides a quantitative measure of the semantic coherence of a topic. The second metric is topic diversity [35], which suggests the percentage of unique words in the top 25 words of all topics. Diversity close to 1 means more sundry topics. Following Dieng et al. [35], we define topic quality as the product of topic coherence and topic diversity. Note that topic quality is affected by the topic size, so it makes sense to compare different models on the same layer. As shown in Fig. 3, Gaussian SawETM performs better compared to SawETM, which can be attributed to the Gaussian-distributed embeddings that capture the semantic uncertainties of topics. Incorporating the hierarchical prior knowledge, TopicNet further achieves a higher score than Gaussian SawETM. However, it gets a lower score compared to DLDA-Gibbs in the first two layers, as the layer goes deeper, TopicNet performs better. This is easy to understand since information from data reduces slightly in the shallow layers but severely in the deep layers.

**Visualisation of embedding space:** First of all, we use a 3-layer Gaussian SawETM with 2-dim embedding trained on 20NG for the Gaussian-distribution embedding visualization. The top 4 words from the $16^{\text{th}}$ topic at $1^{\text{th}}$ layer are visualized in Fig. 4(a). As we can see, the words under the same topic are closer in the embedding space, which demonstrates the learned embeddings are semantically similar. Apart from the semantics of words and topics, the shadow range describes the uncertainties and reflects the abstraction levels. Secondly, we visualize the means of Gaussian-distributed embeddings learned by TopicNet with T-SNE [40]. As shown in Fig. 4.2, in the shared embedding space, the words with similar semantics are close to each other, meanwhile they all locate near the topic that has similar semantics to them, which means the TopicNet effectively captures the prior semantic information and discovers more interpretable topics.

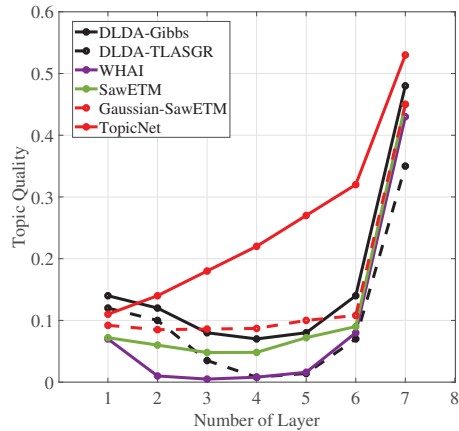

Figure 3: Topic quality as measured by normalized product of topic coherence and topic diversity (the higher the better) with the varied number of layers. Each layer's topic quality is influenced by its layer size, which is set as [411, 316, 231, 130, 46, 11, 2] from the bottom layer to the top layer.

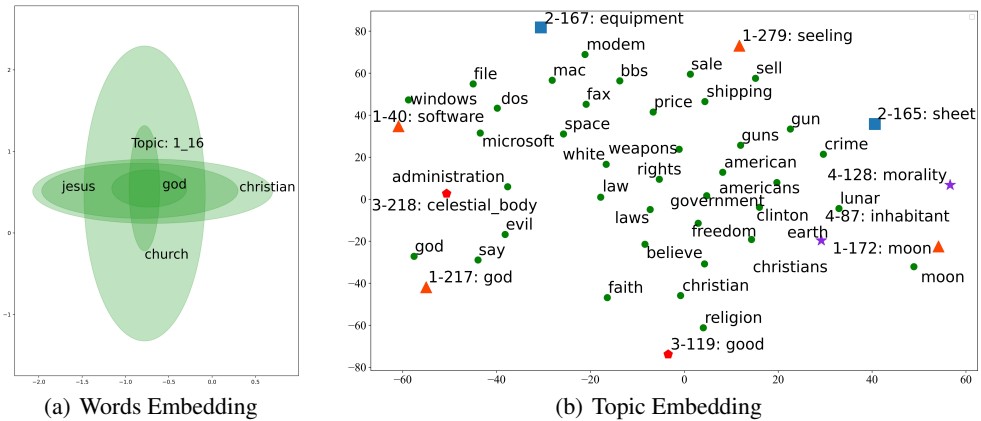

(a) Words Embedding  (b) Topic Embedding

Figure 4: (a) 2-dim Gaussian embedding in Gaussian SawETM, which we choose the top four words for the topic and (b) T-SNE visualisation of the mean of Gaussian distribution embedding in TopicNet. (The Topic: `t-j : xx` denotes the $j^{th}$ topic at `t`$^{th}$ layer and `xx` represent the concept about the topic.)

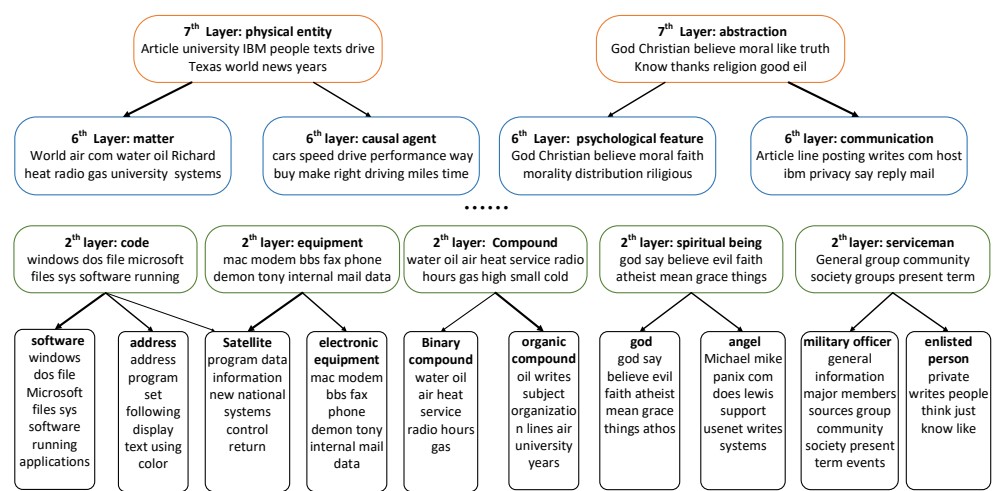

Figure 5: An example of hierarchical topics learned by a 7-layer TopicNet, we only show example topics at the top two layers and bottom two layers. Boldface indicates the concept of each topic.

**Hierarchical structure of TopicNet:** Distinct from many existing topic models that rely on post-hoc visualization to understand topics, we can easily understand the semantic of the whole stochastic network via the predefined TopicTree. Specially, the hierarchical structure of TopicNet is shown in Fig. 6. The semantic of each topic can be represented by the concept in the TopicTree, which is more interpretable compared to traditional topic models. This result further confirms our motivation described in Fig. 1.

## 5  Conclusion

To equip topic models with the flexibility of incorporating prior knowledge, this paper makes a meaningful attempt and proposes a knowledge-based hierarchical topic model — TopicNet. This model first explores the use of Gaussian-distributed embeddings to represent topics, and then develops a regularization term that guides the model to discover topics by a predefined semantic graph. Compared with other neural topic models, TopicNet achieves competitive results on standard text analysis benchmark tasks, which illustrate a promising direction for the future development of text analysis models toward better interpretability and practicability.

## Acknowledgments

Bo Chen acknowledges the support of NSFC (U21B2006 and 61771361), Shaanxi Youth Innovation Team Project, the 111 Project (No. B18039) and the Program for Oversea Talent by Chinese Central Government.

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
