# Appendix for TopicNet: Semantic Graph-Guided Topic Discovery

## A   Detailed discussion of our work

### A.1   Limitations

This paper proposes a novel knowledge-based hierarchical topic model called TopicNet, which can inject prior knowledge to guide topic discovery. In particular, Gaussian SawETM is first proposed as a common hierarchical topic model, which represents both words and topics by Gaussian-distributed embeddings and builds the dependency between different layers by the Sawtooth Connector module [14]. Experiments show that Gaussian SawETM, which has the ability of capturing semantic uncertainty, gets better performance compared with SawETM, especially for the document classification task. However, Gaussian SawETM requires higher memory for training, especially for datasets with big vocabulary size. Due to the high dimension of intermediate variables (e.g. $50000 * 100 * 256$) in computing the sum of two matrix (e.g. $50000 * 100 * 1$ and $1 * 100 * 256$), the gradient of the intermediate variable requires large storage. While the experiments about topic discovery demonstrate that TopicNet can express the semantics of each node in the network by concepts, as an exploratory model, TopicNet guided by a pre-defined graph may not provide better performance for certain specific quantitative tasks.

### A.2   Broader impact

The proposed TopicNet can be used for text analysis, such as topic discovery and mining hierarchical document representation. Distinct from traditional topic models, TopicNet can express the meaning of each node in the network by pre-defined concepts, which shows better interpretability. With its interpretable latent features, we can further understand the behavior of the model instead of just knowing a result, this can be attractive and important in some special applications. For instance, to recommend articles with specific topics to users, it is necessary to understand the user's interest and incorporate it as a priori into the model to provide a reasonable recommendation. Users could also try to understand why a certain recommendation has been made by the proposed model, which results in more trust.

There is significant recent research interest in pre-trained language models, such as BERT [41], GPT2 [42], which are fine-tuned to achieve the state-of-the-art performance in a variety of natural language processing tasks. Despite their promising performance, many researchers tend to strongly rely on the numerical performance but pay less attention to their interpretability. We hope our work can motivate machine learners to focus more on the study of understanding the model.

## B   Detailed figure of the proposed model

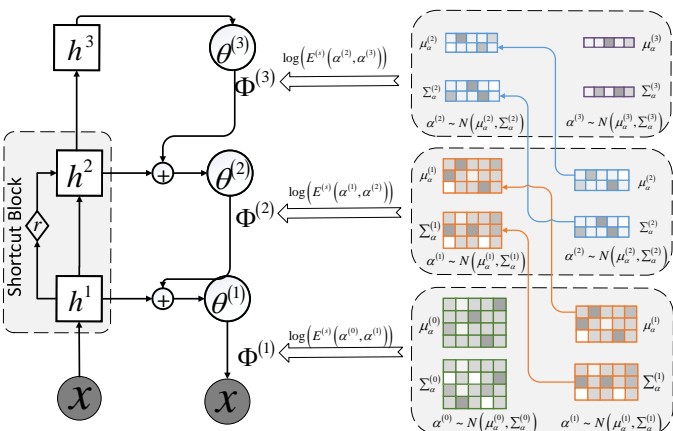

Figure 6: Detailed figure of Gaussian SawETM

## C   Implementation details

**PPL experiments:**   We set the embedding size as 100 and the hidden size as 256. For optimization, the Adam optimizer is utilized here [43] with a learning rate of 0.01. We set the size of minibatch as 200 in 20NG datasets while 2000 in RCV1 and Wiki datasets. All experiments are performed on Nvidia GTX 8000 GPU and coded with PyTorch as attached in the supplement. For the models in the DLDA group and DNTM group, the network structures of 15-layer models are [256, 224, 192, 160, 128, 112, 96, 80, 64, 56, 48, 40, 32, 16, 8], while the network size is set as 256 for the methods in LDA group. Due to the network structure of TopicNet is defined by the TopicTree and the main task of TopicNet is topic discovery, so we do not compare TopicNet with traditional topic models on PPL experiments.

**Document classification and Clustering experiments:**   We set the embedding size as 50 and the hidden size as 256. The 20NG dataset is used with a vocabulary of size 20,000. The network structure is set as [416, 89, 12, 2] for the 20NG dataset and [264, 66, 11, 2] for the R8 dataset.

**Topic quality and visualization experiments:**   We use the 20NG dataset with a vocabulary of size $2,000$. The embedding size is set as 50 and the hidden size is set as 1000. The network structure is set as [411, 316, 231, 130, 46, 11, 2]. The other settings are same with previous experiments.

## D   The details of building TopicTree

Hierarchical topic models, such as gamma belief network, is structured as a tree where all the leaf nodes of a parent node are on the same floor. Due to the complexity of language, the structure of WordNet [16] is a directed graph but not a tree [44]. So to inject the prior knowledge from WordNet to topic models naturally, we need to first construct a TopicTree that have similar structure with gamma belief network.

**Top-down traversal:**   For a $T$-layer TopicTree, the original structure for the top $T$ layers in the WordNet is kept. Due to the bottom layer concept is defined as the word layer in the topic model, all the children nodes of the bottom layer node concept are connected to its ancestor node. For example, as shown in Fig. D, we construct a 4-layer TopicTree from the WordNet. At first, the top $t$ layers are kept. Then, the children node "dog" of node "mammal" is connected to the parent node "animal." The node "male" is the same setting. After this process, we can get a TopicTree as shown in Fig. D (b). Note that, this Tree is built for all the concepts in WordNet, while the vocabulary of the special dataset does not have all the concepts. So to adapt to the domain of the dataset, we need to construct a TopicTree for the special dataset.

**Down-top traversal:**   Given the TopicTree as shown in Fig. D (b), we need to build a new TopicTree to adapt the domain of a special dataset. Specially, the bottom layer concept is set as word layer, the intersection of the dataset vocabulary, and then traversing the parent node from the bottom layer to top layer to construct a new TopicTree.

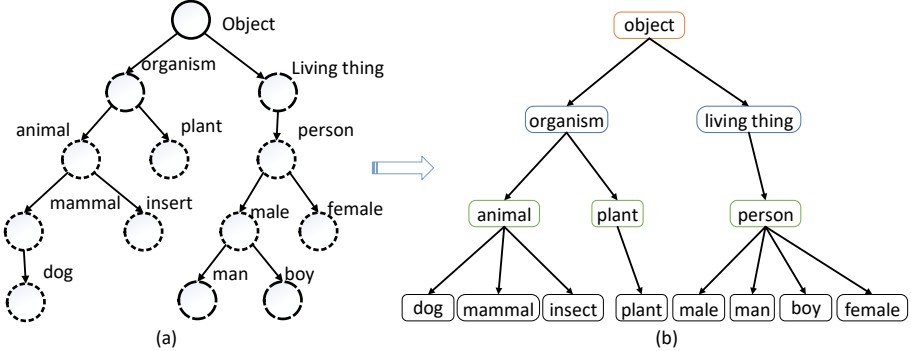

Figure 7: An example of (b) 4 layer TopicTree built from (a) WordNet.

Table 4: Comparison of testing time.

| Methods | Depth | Time(s) |
|---|---|---|
| PGBN [6] | 1 | 9.74 |
| PGBN [6] | 5 | 11.25 |
| Gaussian SawETM | 1 | 0.62 |
| Gaussian SawETM | 5 | 0.85 |

Table 5: Comparison of document classification and clustering performance.

| Methods | Depth | Classification | Clustering | |
|---|---|---|---|---|
| | | | ACC | NMI |
| LDA [1] | 1 | 62.1±0.4 | 37.4±0.4 | 38.1±0.3 |
| NVITM [38] | 1 | 61.3±0.3 | 40.2±0.3 | 41.2±0.2 |
| ETM [35] | 1 | 61.6±0.3 | 39.3±0.3 | 38.4±0.4 |
| PGBN [6] | 4 | 63.2±0.5 | 38.4±0.2 | 39.2±0.3 |
| WHAI [25] | 4 | 62.5±0.4 | 38.5±0.2 | 37.2±0.3 |
| SawETM [14] | 4 | 62.0±0.3 | 41.6±0.4 | 38.4±0.2 |
| Gaussian SawETM | 4 | 62.8±0.3 | 42.3±0.2 | 39.0±0.2 |
| TopicNet | 4 | 65.0±0.3 | 43.2±0.2 | 42.2±0.4 |

# E   Other experiment results

**Test time results**   For the perplexity evaluation, the DLDA group requires iteratively sampling to infer latent document representations in the test stage, despite getting better performance. To support this argument, the test time results (average seconds per document) on 20Newsgroup dataset are shown in Tab. 4.

**Document Classification & Clustering on 20NG dataset with a vocabulary size of 2000:**   We construct the semantic graph from WordNet in all experiments/datasets. Indeed, the quality of the constructed semantic graph has a great impact on the clustering/classification performance. To further illustrate the effectiveness of the proposed model, we build a better-fitted semantic graph for the 20Newsgroups dataset with a vocabulary size of 2000. Choosing a vocabulary of 2000 terms (words) for the 20 newsgroups, we compare the 2000 terms to these in the WordNet, which consists of over 70,000 terms, and find 736 terms that are shared between the 20newsgroups and WordNet. Extracting a four-layer subgraph from the WordNet that are rooted at these 736 terms, we obtain a prior WordNet with [736,314,152,52,11] nodes from the bottom to top layers. The experiment's results on 20Newsgroup dataset with a vocabulary size of 2000 are summarized in Tab. 5. The results show that TopicNet achieves better performance compared with baseline models that do not inject prior knowledge, which confirms our motivation. For Tab. 2, we choose a vocabulary of 20,000 terms, which has 4693 terms that overlap with the vocabulary of the WordNet. Using these 4693 shared terms, we extract a [4693,496,89,11,2] subgraph as the prior WordNet. With not only a larger vocabulary size, but also a wider first topic layer (496 topics instead of 314 topics), it is hence not surprising that a better performance can be achieved for downstream tasks.