# OpenReview forum: "TopicNet: Semantic Graph-Guided Topic Discovery"
_NeurIPS.cc/2021/Conference — NeurIPS 2021 Poster_

### Official Review · Reviewer_Thqc · 2021-06-24

**Rating:** 6
**Confidence:** 3

**Summary:**

The authors propose a topic model that aim at embedding words and topics into a latent space via Gaussian embeddings. Additional constraints in the training procedure allows to incorporate prior knowledge on the topics, so that they are able to include semantic topic relations.

The model is evaluated on a large set of public datasets and on a number of tasks. It achieves better results with respect to baselines on clustering tasks, and comparable on classification tasks. However, the main advantage is the increase interpretability of topics and topic hierarchy.

**Main Review:**

The paper is sometimes quite difficult to read. However, the authors made a clear effort to be exhaustive in explaining mathematical details and experiment settings.

I am not sure I understand how it is possible to incorporate additional knowledge on the topic semantics. The authors mention that "Since the Gaussian SawETM projects the topics of all layers to a shared embedding space, such a goal can be achieved by imposing constraints on the topic representations in the shared embedding space.". However, as each topic is a collection of words, and the topic is embedded in a latent space, the semantic of the collection of words is lost. How the topic semantic (if the topic is automatically discovered by the procedure) can be mapped to the external knowledge graph between the entities?

The authors do not mention the scalability of the model. One main limitation of standard topic model is their scalability when increasing the vocabulary size, as for each topic all of the vocabulary is needed to be sampled to compute the probability distribution. From the proposed approach, it seems that the method aim at embedding topics into a latent space. It would be nice to assess the scalability in terms of words or topics.

In terms of experiments, the authors mention that they evaluated the model also in a classification task. However, it's not clear to me what the classification task is as the dataset they describe are textual documents. What labels have the document in the datasets? Each document has a "category" as a label?

Highlighting the best results in Table 2 would make the table easier to read. It seems that in general the proposed method is better than the baselines on the various clustering tasks, and slightly worse for classification tasks. When adding prior knowledge, the method has actually worse performance. The authors mention that "TopicNet does not perform better maybe because the prior information is irrelevant with the document classification and clustering tasks". However, the additional information on topic relation should better guide the model, and I am a bit surprise this is happening. As I haven't fully understood how to include the prior information in the model I think this is an important point to explore better.

Minor:
typo: negetive (page 6)

**Time Spent Reviewing:**

2.5

---

> ### Author Response · Authors · 2021-08-10
> **Detailed response to Reviewer 4's questions**
>
> We appreciate your comments and suggesions. The concerns have been addressed as below.
>
> **Q1:** How the topic semantic (if the topic is automatically discovered by the procedure) can be mapped to the external knowledge graph between the entities?
>
> **A1:** In traditional topic models, each topic is a collection of words, with different words having different proportions. While in Gaussian SawETM, all topics and words are projected into a shared embedding space, where each word or topic is represented as a Gaussian distribution so that a semantically more general topic (corresponding to a broad distribution) could encompass many specific words with similar semantics (corresponding to many narrow distributions). Due to this point, we can utilize the external knowledge graph to assign a semantic concept to each topic, and then guide the model learning through a regularized loss (encouraging that the distributions of semantically similar topics and words are close while the distributions of general topics encircle the distributions of specific words as much as possible).
>
> **Q2:** It would be nice to assess the scalability in terms of words or topics.
>
> **A2:** Good advice. We will add the assessment of the scalability in terms of words or topics
>
> **Q3:**  What labels have the document in the datasets? Each document has a "category" as a label?
>
> **A3:** Our classification task is based on 20 Newsgroups and R8, where each document has a "category" as a label. For the 20 Newsgroups (or R8) dataset, our goal is to classify each document into one of the 20 (or eight) different newsgroups.
>
> **Q4:** Highlighting the best results in Table 2 would make the table easier to read. It seems that in general the proposed method is better than the baselines on the various clustering tasks, and slightly worse for classification tasks. When adding prior knowledge, the method has actually worse performance. The authors mention that "TopicNet does not perform better maybe because the prior information is irrelevant with the document classification and clustering tasks". However, the additional information on topic relation should better guide the model, and I am a bit surprise this is happening. As I haven't fully understood how to include the prior information in the model I think this is an important point to explore better.
>
> **A4:** Actually, a more effective topic hierarchy based on the dataset will help improve the clustering or classification performance, this could be demonstrated by our supplementary experiments (please read the A2 in the reponse to reviewer 3). However, such dataset-specific prior knowledge is often not available, we therefore seek to construct the semantic graph from some general knowledge graphs like WordNet, based on the vocabulary of different datasets. As a result, the semantic graph constructed by this way somewhat deviates from the dataset, which potentially affects the clustering/classification performance of TopicNet.
>
> We will address all remaining minor suggestions in the final revision.

---

> > ### Comment · Reviewer_Thqc · 2021-08-17
> > **Comment to response**
> >
> > I thank the authors for the clarification and for answering my questions.
> >
> > I think the paper proposes a relevant and interesting contribution because of the possibility of adding prior knowledge in the form of a semantic graph to topic models to guide the inference. However, by also looking at the reviews from others, I am not completely convinced by the status of the experiments and validation of the method. I appreciate the fact that the main strength of the proposed method is the interpretability when having a semantic network appropriate for the task at hand. But, as the authors mention, it is often not available in practice, and one need to resort to a general semantic network that may not be optimal for all downstream tasks. Therefore, its use is conditioned on which task is more important: interpretability or accuracy. It would be beneficial to showcase at least a very specific experiment where, by adding the prior knowledge, not only the method would improve results' interpretability (which is expected given the semantic network), but also better topics and performance on downstream tasks.
> >
> > While there is no explicit assessment on scalability at this stage, by also looking at the supplementary material, it seems that the model requires high memory for training. As the authors mention, there is the possibility that the method could not be used for certain quantitative tasks, which leaves my doubts in place regarding how easy and how beneficial would be the implementation and the use of the method in practice.
> >
> > I think my rating is in line with my current assessment. Therefore, I maintain my rating at this stage.

---

> > > ### Author Response · Authors · 2021-08-17
> > > **Response to new official comment**
> > >
> > > Thank you very much for your additional feedback. We appreciate your agreeing with our contribution in providing the possibility of adding prior knowledge in the form of a semantic graph to guide the learning of hierarchical topics.
> > >
> > > Your remaining concern is whether the provided experiments are sufficient to validate that point. In particular, one may need to make an uneasy decision between interpretability and accuracy when deciding whether to use TopicNet. We appreciate your highlighting that concern and suggesting a specific experiment to try, which will be taken into careful consideration when revising the paper. We believe the provided possibility of injecting specific prior knowledge embedded in graphs would open the door to many interesting applications.
> > >
> > > We acknowledge that our current PyTorch code included in the supplement has not yet been optimized for memory and speed (the current bottleneck is not the model size but the large memory requirement for gradient backpropagation brought by the use of Gaussian embeddings, especially when the vocabulary size is as large 40000), which is the reason we have not tried TopicNet with many layers (please also see “Additional clarifications to address the remaining doubts” to Reviewer 93MZ). We are continually working on improving the code, in particular, finding memory-efficient ways to process a very wide and deep graph introduced as a prior guidance under a large-size vocabulary. We are confident we will be able to significantly speed up the code for TopicNet and reduce its memory requirement in the near future.

---

### Official Review · Reviewer_aZKr · 2021-07-16

**Rating:** 6
**Confidence:** 4

**Summary:**

This paper mainly proposes a new hierarchical topic discovery framework that is guided by prior (tree-structured) knowledge. The construction of the framework is broken down into two steps: Modeling semantic hierarchy with a Gaussian-embedding-based topic model (Gaussian SawETM), and injecting the prior semantic knowledge by regularizing via entailment relationship. The model is evaluated on a set of benchmarks with perplexity, topic quality, and clustering/classification protocols. Qualitative studies are also shown to demonstrate the interpretability of the model.

**Limitations And Societal Impact:**

I feel that the authors have addressed the limitations in a reasonable way.

**Main Review:**

[Originality]

Overall, the novelty is at the borderline of a NeurIPS paper. In Sec. 2, the Gaussian SawETM model is essentially a combination of SawETM and Gaussian embeddings, so the ideas are not new there. In Sec. 3, the injection of prior knowledge is implemented as a set of regularization over the entailment relationship in the semantic graph (like WordNet), and the modeling of the transitivity and antisymmetry is commonly seen in hierarchy modeling studies (the line of hyperbolic embeddings), thus the design is reasonable but not particularly providing new insights. However, I like the overall motivation of this paper to guide topic discovery with prior (graph-structured) knowledge, which is not commonly seen in the topic modeling literature.

[Quality]

The Gaussian SawETM model borrows the idea from a recent topic modeling paper SawETM to alleviate the model collapse issue for deep hierarchical topic models, and further leverages Gaussian embeddings to model semantic uncertainties of words. The model inference follows standard VAE-based neural topic models for optimizing the ELBO. The TopicNet framework, built upon the Gaussian SawETM model, incorporates prior tree-structured knowledge by regularizing the Gaussian embeddings to align with the partial order of semantic concepts as a new objective added to the ELBO objective. The overall design of the model is principled and neat.

The empirical evaluation is conducted on a set of benchmarks with metrics commonly used in topic modeling evaluation (perplexity, topic quality, and clustering/classification). The proposed models (Gaussian SawETM & TopicNet) demonstrate comparable or better performance than existing neural topic models. So the evaluation also seems convincing to me.

[Clarity]

The paper is overall clearly written and well-organized. The method sections clearly draw connections with related studies and motivate each component. The experiment section clearly states the setup/evaluation protocol and presents detailed analyses.

[Significance]

The significance of this paper is probably at an okay level--although the proposed models outperform baselines in some tasks (clustering/classification), they are not better under perplexity evaluation. Also, the guided version (TopicNet) underperforms the unsupervised version (Gaussian SawETM) for clustering/classification, which is inconsistent with the major motivation of the paper. However, I appreciate the overall direction of incorporating prior semantic knowledge into topic discovery, and feel that the paper has made an interesting exploration overall.

[Questions]

I hope the authors could clarify the following points:
* Is WordNet used as the semantic graph in all experiments/datasets? Could you use other prior structures that better fit the specific domains/datasets? It is mentioned in the document clustering/classification experiments that TopicNet does not perform better than Gaussian SawETM because the prior information is irrelevant with the document classification and clustering tasks. This observation largely weakens the motivation of this paper that some prior guidance can help the topic discovery process.
I'm curious whether a manually curated topic hierarchy based on the dataset will help improve the clustering/classification performance--even that will make the comparisons with baselines unfair, but at least that will confirm the motivation and demonstrate the effectiveness of TopicNet on modeling the given semantic graph.
* It seems to me that the current setup of TopicNet can only model tree structures. If this is true, calling the model "Semantic Graph-Guided" might be a bit inappropriate, as the tree structure is only a subset of graph structures.
* It is mentioned in the perplexity evaluation that the DLDA group requires iteratively sampling to infer latent document representations in the test stage, despite getting better performance. I feel that some concrete run time reports are necessary to support this argument (are they really that costly compared to the DNTM group?).

[Summary]

Pros:
* The paper is clearly written and well-organized; the incorporation of prior semantic knowledge is well-motivated.
* The overall framework is reasonable and principled.
* The evaluation is comprehensive.

Cons:
* The novelty is not that high, as many essential components are borrowed from previous work.
* The empirical evidence that TopicNet underperforms Gaussian SawETM in applications raises concerns about the basic assumption/motivation of the paper.

################

Updates after rebuttal:
I would like to thank the authors for conducting new experiments and providing clarifications that addressed my concerns. I increased my rating from 5 to 6.

**Time Spent Reviewing:**

8

---

> ### Author Response · Authors · 2021-08-10
> **Detailed response to Reviewer 3's concern on motivation and model's  effectiveness**
>
> We appreciate your detailed comments and suggestions. They identify some important points which we hope to clarify and address here and in our revision.
>
> **Concern** on originality and novelty.
>
> **Answer:** please read the anwser about “Concern on originality” in the response to Reviewer s9bs.
>
> **Q1:** I'm curious whether a manually curated topic hierarchy based on the dataset will help improve the clustering/classification performance--even that will make the comparisons with baselines unfair, but at least that will confirm the motivation and demonstrate the effectiveness of TopicNet on modeling the given semantic graph.
>
> **A1:** We construct the semantic graph from WordNet in all experiments/datasets. Indeed, the quality of the constructed semantic graph has a great impact on the clustering/classification performance. For example, we build a better-fitted semantic graph for the 20Newsgroups dataset with a vocabulary size of 2000. We conduct the same clustering/classification experiment, the results (see the following table) show that TopicNet achieves better performance compared with baseline models, which confirms our motivation. The 20Newsgroup dataset experiment’s results, with a vocabulary size of 2000, are summarized as bellow:
>
> |     Model    | #Layers | Classification acc | Clustering acc | Clustering NMI |
> |:------------:|:-------:|:------------------:|:--------------:|:--------------:|
> |      LDA     |    1    |        62.1        |      37.4      |      38.1      |
> |     NVTM     |    1    |        61.3        |      40.2      |      41.2      |
> |      GBN     |    4    |        63.2        |      38.4      |      39.2      |
> |     WHAI     |    4    |        62.5        |      40.7      |      38.6      |
> |    SawETM    |    4    |        62.0        |      41.6      |      38.4      |
> | Gauss SawETM |    4    |        62.8        |      42.3      |      39.0      |
> |   TopicNet   |    4    |        65.0        |      43.2      |      42.2      |
>
> **Q2:** It seems to me that the current setup of TopicNet can only model tree structures. If this is true, calling the model "Semantic Graph-Guided" might be a bit inappropriate, as the tree structure is only a subset of graph structures.
>
> **A2:** In our experiments, we do use the tree structure, but actually TopicNet can also handle more complex structures such as directed acyclic graphs, where a child node can have multiple parent nodes.
>
>
> **Q3:** It is mentioned in the perplexity evaluation that the DLDA group requires iteratively sampling to infer latent document representations in the test stage, despite getting better performance. I feel that some concrete run time reports are necessary to support this argument (are they really that costly compared to the DNTM group?).
>
> **A3:**  The test time results (average seconds per document) on 20Newsgroup dataset are as follows:
>
> |              |   Layer |  Time   |
> |:------------:|:-------:|:-------:|
> |  DLDA-Gibbs  |    5    | 11.25   |
> |     DNTM     |    5    |  0.85   |
>
> we will give detailed test time results on the different datasets and different network structures in the revision.

---

> > ### Comment · Reviewer_aZKr · 2021-08-24
> > **Thanks for adding the new results**
> >
> > I would like to thank the authors for adding the new results. I appreciate the new clustering/classification results in A1, as well as the concrete run time report.
> >
> > However, I would like to obtain further clarifications regarding the new results:
> > * Could you share the better-fitted semantic graph you built for the 20Newsgroups dataset (not necessarily the complete graph, but some example cases to illustrate what words, when used as the prior guidance, are likely to be helpful for which topics)?
> > * Could you explain why the new results in the A1 Table are lower than the corresponding numbers in Table 2 of the original paper?

---

> > > ### Author Response · Authors · 2021-08-24
> > > **Further clarifications**
> > >
> > > We sincerely appreciate your additional feedback. We are glad to provide clarifications to your two questions.
> > >
> > >   - **A1:** Choosing a vocabulary of 2000 terms (words) for the 20 newsgroups, we compare the 2000 terms to these in the WordNet, which consists of over 70,000 terms, and find 736 terms that are shared between the 20newsgroups and WordNet. Extracting a four-layer subgraph from the WordNet that are rooted at these 736 terms, we obtain a prior WordNet with [736,314,152,52,11] nodes from the bottom to top layers. We show six example topics (out of 314 nodes from the bottom topic layer) and their corresponding children nodes (out of 736 nodes from the bottom word layer) in the following table, which illustrates how a particular concept is being represented in the 20 newsgroups. For example, the “military_action” concept, which has “assault”, “defense”, and “war” as its children nodes, has become tightly connected to the Waco Siege heavily discussed in this dataset; the “macromolecule” concept, which has “oil” as its single child node, has been represented by a topic related to vehicle, which is interesting as the 20newsgroups dataset has the following two newsgropus--- “rec.autos’ and “rec.motorcycles”---but not a newsgroup clearly related to biology and chemistry; and the “Atmosphere” concept, which has “Sky” as its single child node, has been represented by a topic related to NASA, reflecting the fact that “sci.space” is one of the 20 news groups.
> > >
> > > | Bottom layer topic  | Concept       | Children word nodes         | Topic words |
> > > | -------------    |:-------------:|:-------------:|:-------------:|
> > > | 11               | military_action     | assault defense war   |gun fbi guns koresh batf waco assault children compound weapons|
> > > | 131              | coding_system     | address shareware software unix windows|windows dos driver microsoft drivers running applications unix using network|
> > > | 178              | macromolecule   | oil|car bike cars engine oil saturn ford riding miles road |
> > > | 186              | polity     | government|government law rights state police right federal shall crime court laws|
> > > | 296              | atmosphere    |sky|space launch nasa gov satellite earth mission shuttle hst orbit sky|
> > > |298               | color     |black blue brown color green red|color green blue red black brown led subject showed lines|
> > >
> > >
> > >  - **A2:** For Table 2 of the original paper, we choose a vocabulary of 20,000 terms, which has 4693 terms that overlap with the vocabulary of the WordNet. Using these 4693 shared terms, we extract a [4693,496,89,11,2] subgraph as the prior WordNet. With not only a larger vocabulary size, but also a wider first topic layer (496 topics instead of 314 topics), it is hence not surprising that a better performance can be achieved for downstream tasks.
> > >
> > > - We would also like to clarify that the full WordNet has 18 layers. In this paper, to extract a subgraph, we first determine the depth of the subgraph as $L$, we then flatten the WordNet below the top $L$ or $L+1$ layers. A $L$ subgraph is then extracted from this partially flattened WordNet. We believe there are better ways to extract prior WordNet subgraphs to guide the topic discovery, which is not the focus of this paper and hence left for future study.

---

> > > > ### Comment · Reviewer_aZKr · 2021-08-25
> > > > **Thank you for the new results and the clarifications**
> > > >
> > > > I would like to thank the authors for conducting new experiments and providing clarifications that addressed my concerns. The new results demonstrated that with a properly chosen semantic tree/graph that better fits the corpus, the discovered topic quality and clustering/classification performance could be improved. This validated the motivation of this paper and the effectiveness of the proposed approach. Overall, I feel that the paper makes interesting contributions towards incorporating prior knowledge into topic discovery (this line of study is not commonly seen), and I appreciate the principled approaches, as I indicated in my initial review. Based on the above considerations, I increased my rating from 5 to 6.

---

### Official Review · Reviewer_93MZ · 2021-07-16

**Rating:** 7
**Confidence:** 4

**Summary:**

The paper builds on the recently proposed SawETM hierarchical topic model and combines it with a knowledge graph so that each node in the knowledge graph corresponds to a topic. They change it so that every topic is embedded with a Gaussian distribution. This enables to enforce a hierarchical ordering constraint on the topics. Experiments show that the model discovers deeper, more interpretable topics and finds better document representations.

**Ethical Concerns:**

As I wrote in the main review:
The example in Fig. 1a) could be read as being sexist, as mother, son and home are in one topic, but father is not. I would choose a different example or explain how this sort of bias can be prevented using the prior knowledge possibly.

**Ethics Review Area:**

["Discrimination / Bias / Fairness Concerns"]

**Limitations And Societal Impact:**

One main limitation from my point of view is the fact that we do not know if the knowledge graph is appropriate for the dataset, as I wrote in the review:
How can we be sure that the knowledge graph has anything to do with the dataset that we use? What if additional topics are present or the topics in the knowledge graph are more detailed than the dataset calls for?

So the negative impact could be that topics in the dataset are not found because they are not present in the prior knowledge. Limitations are not really addressed in the paper.

**Main Review:**

This paper presents an interesting idea of incorporating knowledge graphs with hierarchical topic models. My criticism is limited to explanation and inconsistency in experiments. The article proposes two novel methods for Hierarchical Topic Modeling. Gaussian SawETM - A probabilistic generative model to capture semantic uncertainties of words and topics. TopicNet - A knowledge-based hierarchical topic model which incorporates prior knowledge (belief) of the real world. TopicNet employs Gaussian SawETM and learns the hierarchies based on prior beliefs such as knowledge graphs. The idea is theoretically sound and has a mathematical
explanation. The structure and quality of explanation can be improved in some
sections. More consistent experiments and comparisons would have validated
what the authors claim, the absence of which can leave a reader curious.

The quality of the explanation is inconsistent throughout the article. While the
Introduction section is not properly structured, it does a good job of explaining
the content it provides with the help of diagrams. Whereas, Section 2 and 3,
which form the proposed model, lack quality of explanation.
It has the following explanation problems:

- No proper figure is given for the proposed TopicNet model.
- The figure provided for Gaussian SawETM is a general Gamma Belief Net-
work, it has no representation of upward and downward encoder networks
as described in Section 2.3.
- The authors claim to have reduced mode collapse by their use of Gaussian
SawETM, but provide inconsistent results in Table 1 and Table 2. In perplexity representation depth till 15 are accounted for but in classification
depth of only 1 and 4 are accounted for. The depth inconsistency further
continues in Figure 4, where the embedding visualization is done for Gaussian SawETM with a depth of 3. Depth should be made consistent while
comparison in all tables as well as figures, since claims based on different
layers, coincide with results of different layers.
- While presenting the Topic Quality, the authors themselves hint at the
reason behind the exceptional performance of TopicNet on severe infor-
mation reduction from data in the deep layers. This specific case will lead
to more diversity in words, hence Topic Quality can be a slightly skewed
metric here for the judgement. Results of Topic Coherence based on NPMI
can also be presented to clear the doubt.
- Only selected hierarchies are presented. Human evaluation measures such
as word intrusions can be utilized to evaluate the hierarchies.


Two further questions I have for the authors are:

- How can we be sure that the knowledge graph has anything to do with the dataset that we use? What if additional topics are present or the topics in the knowledge graph are more detailed than the dataset calls for?
- How are the semantic concepts themselves injected into the topic model? I only see how the relationships between the concepts are injected. But how do you make sure that the topic tree node "animal" actually corresponds to a topic with words related to animals?


Minor comments:
- Related work section and introduction are mixed together, I would suggest to separate them into two separate parts.
- The heading of Section 2 should maybe better be changed to "Gaussian SawETM"
- In some cases such as [7], the arXiv version of a paper is cited even though it is published in a peer-reviewed conference.
- The references are very inconsistent, e.g. compare [25] and [26]
- The example in Fig. 1a) could be read as being sexist, as mother, son and home are in one topic, but father is not, so only mothers walk their son to school?
- the example in Fig. 1a) seems not very meaningful, for example the topic "road read bike walk", makes not much sense, as read has nothing to do with road.
- line 128 BOund-> bound
- Figure. 5 -> Fig. 5
- Figure. 1 -> Fig. 1
- tpoic -> topic
- 7-ayer -> 7-layer
- negetive -> negative



*************************
my comments were sufficiently addressed and I am increasing my rating

**Needs Ethics Review:**

Yes

**Time Spent Reviewing:**

4

---

> ### Author Response · Authors · 2021-08-10
> **Detailed response to Reviewer 2's concern on the quality of the explanation and questions**
>
> We appreciate your comments and suggestions. The concerns have been addressed as below.
>
> **Concern** on the quality of the explanation is inconsistent throughout the article.
>
> **Answer:** For the problem of inconsistent explanation, we will make some appropriate modifications in the final revision according to the reviewer’s advice, including:
>
> 1. We will add a more detailed figure to illustrate the proposed Gaussian SawETM and TopicNet model in the revision, which include upward and downward encoder networks, Sawtooth factorial Gaussian embedding GBN, and prior knowledge module.
>
> 2. For comparison that includes TopicNet, the biggest depth depends on TopicNet, since in TopicNet the depth of topic model should correspond to the depth of semantic graph. However, constructing a very deep (depth=15) semantic graph will lead to a large topic size (10000) in the bottom layer, which cannot be handled with our computing sources.
>
> 3. Overview, we will add the topic coherence metric based on NPMI in the revision;
>
> 4. Thank you for your suggestion. We leave human evaluations for topic hierarchies for future study.
>
> **Q1:** How can we be sure that the knowledge graph has anything to do with the dataset that we use? What if additional topics are present or the topics in the knowledge graph are more detailed than the dataset calls for?
>
> **A1:** Indeed, it is a critical issue for certain downstream tasks, such as classification and clustering, that the constructed semantic graph should match the dataset. On the other hand, the concepts that are present in the knowledge graph but rarely appear in the dataset due to biases could become very beneficial in correcting these biases. In our experiment, we propose an algorithm to build a semantic graph for each dataset from WordNet (The detail is described in Appendix C). Specifically, we build a semantic graph based on the dataset’s vocabulary, which makes the constructed graph match the dataset as much as possible. However, this is not a perfect method and some potential solutions such as the finetuning or pruning of the semantic graph may be helpful. Note that it is difficult and not the focus of our paper, we leave it as a possibility for future work.
>
> **Q2:** How are the semantic concepts themselves injected into the topic model? I only see how the relationships between the concepts are injected. But how do you make sure that the topic tree node "animal" actually corresponds to a topic with words related to animals?
>
> **A2:** Let's take the example of a topic tree node "animal".  Specifically, assuming the topic tree node “animal” has a child node “cat”. For one thing, the regularized loss (a contrastive loss) would encourage the embeddings of “cat” to be encircled by the embedding of “animal”. For another, the original ELBO loss would encourage the embeddings of semantically similar words to be close in embedding space, such as “cat” and “dog” . Under the influence of above two factors, the embedding of “animal” will be closer with “cat” and “dog”. Actually, we only establish the semantic connection between the topics and the words at the bottom layer, but this semantic relationship can be passed up through the entailment relationships between concepts.
>
> We thank the reviewer for expressing the ethical concerns about the example we used in Fig. 1a), which might be read as being sexist. We will choose a different example in the final revision. We will also study whether constructing proper prior knowledge graph can be used to help mitigate data biases.
>
> We will fix all typos and address other minor suggestions in the final revision.

---

> > ### Comment · Reviewer_93MZ · 2021-08-16
> > **Clarification is appreciated, however not all doubts cleared**
> >
> > I thank the authors for the clarification regarding the loss function. I am still not entirely clear about why Table 1 goes to a depth of 15 and Table 2 goes to a depth of 4. At the very least, this should be mentioned in the text also.
> > I acknowledge that I have read the response and the "Clarification on the main objective of injecting the semantic graph prior". However, in this clarification I have not found indications how the authors plan to clarify the main objective in the paper. Since all reviewers had doubts on this point, it seems to indicate that the paper needs to be rewritten in order to account for this.
> >
> > Further questions I had when checking the text again:
> > - why use the Weibull approximation when there are better approximations for the Gamma/Dirichlet such as [1]
> > - In Eq. 2, how is $\gamma$ and $c_j$ defined?
> > - are the authors publishing the code? The code of SawETM has not been published yet which also makes me wonder how the authors managed to implement it so fast after the SawETM paper was only published a few weeks ago. It makes it very likely that the authors are probably overlapping with the authors of the SawETM paper. Not publishing their code makes it difficult to verify the results.
> > - In the "Visualisation of embedding space" part, it is mentioned that a 2-dim embedding is used. For the other experiments, the embedding size is not mentioned, I am wondering how one would choose that embedding size.
> >
> > My overall comment is that the experiments and results are fine overall, I see the merits of the method, but I am not convinced at this stage that the paper is in a polished-enough state that enables reimplementation or adoption of the method by other researchers. Therefore, I maintain my rating.
> >
> >
> >
> > [1] Figurnov, Mikhail, Shakir Mohamed, and Andriy Mnih. "Implicit Reparameterization Gradients." NeurIPS. 2018.

---

> > > ### Author Response · Authors · 2021-08-17
> > > **Additional clarifications to address the remaining doubts**
> > >
> > > We sincerely appreciate your additional comments. We hope our response below could help address your remaining doubts.
> > >
> > > *Q1: Why Table 1 goes to a depth of 15 and Table 2 goes to a depth of 4.*
> > >
> > >  A1: Table 1 shows the results of the Gauss-SawETM, which are used to demonstrate the advantages of introducing Gaussian Embeddings into SawETM, for which the topic structure from bottom to top is [256, 224, 192, 160, 128, 112, 96, 80, 64, 56, 48, 40, 32, 16, 8], as shown in Appendix B. By contrast, Table 2 shows the results of TopicNet, which is a Gauss-SawETM guided by a predefined TopicTree extracted from WordNet. This tree could quickly become too large when it goes deeper. For example, the size of a 10-layer TopicTree, built from top to bottom based on WordNet, is about [2, 10, 50, 300, 800, 1500, 2000, 5000, 8000, 10000]. For this reason, to evaluate the proposed TopicNet, we kept the depth at 4, with the pre-extracted TopicTree at the size of [2,12,89,416] for 20News and [2,11,66,264] for R8. To support a much deeper TopicNet, we will either need to solve the memory and speed issue of dealing with a huge TopicTree, or find ways to prune the TopicTree, which are beyond the scope of this paper.
> > >
> > > *Q2: I have not found indications how the authors plan to clarify the main objective in the paper. Since all reviewers had doubts on this point, it seems to indicate that the paper needs to be rewritten in order to account for this.*
> > >
> > >  A2: Given the extra page allowed for the final version, we will be able to incorporate "Clarification on the main objective of injecting the semantic graph prior" into the introduction. The revisions will be mainly for clarifications, as elaborated in detail in our rebuttal, and better highlighting the key motivations behind various components of the proposed models. We are confident that the paper needs no major changes to achieve these goals.
> > >
> > > *Q3: Why use the Weibull approximation when there are better approximations for the Gamma/Dirichlet such as [1]*
> > >
> > > A3: We chose the Weibull distribution for variational inference because it not only provides a good approximation of the gamma distribution, but also is reparameterizable and simple to compute.  We will clarify in the paper that more sophisticated approximations, such as the Implicit Reparameterization Gradients by Figurnov et al. (2018), have the potential to further improve the performance. However, we’d like to point out that these methods are often not as simple as the reparameterizable Weibull approximation that can be easily incorporated into a variational framework. To be more specific, to estimate the gradient of the shape parameter of the gamma distribution $z\sim Gamma(\alpha,1)$, Implicit Reparameterization Gradients require the computation of the gamma cumulative density function, which is the regularized incomplete Gamma function $\gamma(z,\alpha)$ that does not have an analytic expression and hence requires an additional numerical method to approximate its value. In addition to these added complexities in implementation, we have also found this approach to be much slower in computing the gradients, likely due to the need to approximately compute $\gamma(z,\alpha)$ for each latent dimension.
> > >
> > > *Q4: In Eq. 2, how is $\gamma$ and $c_j$ defined?*
> > >
> > > A4: Following WHAI [1], we fix these hyperparameters as $\gamma=1$ and $c_j=1$.
> > >
> > > *Q5: Are the authors publishing the code? … Not publishing their code makes it difficult to verify the results.*
> > >
> > > A5: The code was included in the supplementary material and we will make it publically available in Github after the acceptance of the paper. Our code base is built on top of GBN [2] and WHAI [1], which are publically available.  We have noticed that the code of SawETM, which is also built on top of the code base of GBN [2] and WHAI [1], has recently become available in GitHub.
> > >
> > >
> > > *Q6: In the "Visualisation of embedding space" part, it is mentioned that a 2-dim embedding is used. For the other experiments, the embedding size is not mentioned, I am wondering how one would choose that embedding size.*
> > >
> > > A6: Due to the page limit, the details of the model setup, including the choices of embedding sizes, were deferred to Appendix B. We will clarify that in the revision.
> > >
> > > [1] Hao Zhang, Bo Chen, Dandan Guo, and Mingyuan Zhou. WHAI: Weibull hybrid autoencoding inference for deep topic modeling. ICLR, 2018
> > >
> > > [2] Zhou, Mingyuan, Yulai Cong, and Bo Chen. "Augmentable gamma belief networks." The Journal of Machine Learning Research 17.1 (2016): 5656-5699.
> > >
> > > [3] Adji B Dieng, Francisco JR Ruiz, and David M Blei. Topic modeling in embedding spaces. TACL 2020.

---

### Official Review · Reviewer_s9bs · 2021-07-16

**Rating:** 6
**Confidence:** 3

**Summary:**

(Summary)
This paper proposes a generative model that can learn hierarchical topics that coherently incorporate a structured prior knowledge graph. By introducing Gaussian embeddings to an existing SawETM, a neural topic model where topics in different levels can relate each other, the authors first propose the Gaussian SawETM. Then their TopicNet based on Gaussian SawETM can learn hierarchical topics coherently to an input knowledge-graph. TopicNet is not only capable of learning hypernym-induced topics as hierarchies, but also equipped with rich interpretability.


**Limitations And Societal Impact:**

No problem has been found.

**Main Review:**

(Originality)
Instead of using Gaussian variables as latent codes like other neural topic models based on VAE, the authors combine an idea of using gamma-distributed latent variables from Weibull hybrid inference (by Zhang et al) and SawETM (by Duan et al) for dependencies in two consecutive layers. Therefore the real novelty comes more from an idea of defining topic template as a topic tree and guiding the topics of each layer learned from the Gaussian SawETM by the corresponding semantic layer in the topic tree through imposing constraints on the topic embeddings. Due to such joining nature, one could equally argue that the contribution in modeling is mostly incremental.


(Clarity)
The writing is generally clear, but readers might need to look at previous papers to understand the details of Weibull method and SawETM. Though the previous work doesn’t have to be completely rephrased, providing some degree of background will benefit future readers.


(Significance)
The contribution of the overall paper seems to be more on applicational side as the proposed models rely on combining two successful neural networks. However, the experiments are limited to using either only small vocabulary datasets or too large vocabularies comparing to the number of documents, possibly weakening statistical robustness. Without demonstrating high performance on more realistic datasets (i.e., the number of topics << the number of words << the number of documents), it may be hard to convince potential users.


(Major Comments and Questions)
1) How is the vocabulary curated? Users in topic modeling typically put non-negligible amounts of efforts in vocabulary curation for properly removing both too frequent and too rare terms, which affects final topic quality. It is unclear how the authors curate the vocabulary for their experiments.

2) As you learn a generative model for unsupervised learning, having a semi-synthetic experiment will be useful for verifying true performance. Note that it is a better practice to first learn realistic topic distributions from real text corpora, later sampling various sizes of documents by using the known topic distributions and randomly sampled multinomial topic proportions. Given ground-truth topic distributions and proportions, you will be able to firmly complement the clustering study.

3) Performance of the most standard LDA model could be changed by a big margin based on the choice of hyperparameters as well as inference method such as Collapsed Gibbs Sampling (CGS) or Collapsed Variational Inference (CVI). How would the baseline LDA perform if using learned hyperparameters with CGS? Does the proposed model still significantly outperform LDA when the LDA is optimized as we use as a part of standard protocol?

4) In Section 4.1, the authors said that “SawETM cannot capture the uncertainties of topics”. What is exactly the uncertainty and how is it mathematically defined? Do you have any real example where SawETM fails to capture these uncertainties, whereas Gaussian SawETm captures successfully? Providing at least one qualitative example will benefit the readers even if it is anecdotal.

5) Being explainable and having topic interpretability are not identical in general category. In many cases, Explainable AI (XAI) tries to understand reasoning or inference path where users cannot transparently see how the forecast comes out. Is the topic discovery explainable in an agreeable sense?

6) Topic diversity is named more frequently as inter-topic similarity or dissimilarity. Read some papers by David Mimno will be helpful. In addition, topic specificity (how far from the unigram corpus distribution) can often help quality measurement jointly with those two metrics.

7) WordNet’s hierarchy semantic is mostly limited only to nouns. What happen if some topics are more about actions than entities, thereby being described more by verbs than nouns in their top words?


(Minor Comments)
1) Line 252, from  form


(After Author-Feedback)
Have updated my previous evaluation.


**Time Spent Reviewing:**

3 hours

---

> ### Author Response · Authors · 2021-08-10
> **Detailed response to Reviewer 1's concerns on originality and statistical robustness**
>
> We appreciate your valuable comments and suggestions. The concerns have been addressed as below.
>
> **Concern** on originality.
>
> **Answer:** The marriage of a hierarchical semantic graph with a deep topic model is actually very challenging, inluding the model building and parameter inference, which has not been done before. Our contribution is proposing that the semantic graph can be naturally incorporated with a hierarchical topic model through Gaussian embeddings via the developed efficient inference. For one thing, we introduce the distribution embeddings, instead of point vector,  into a hierarchical topic model to represent topics and words, which brings the capability of measuring both symmetric and asymmetric similarity. For another, the probability distribution enables us to capture the partial order structure contained in the semantic graph through a regularized loss, which effectively injects prior knowledge to guide the topic learning. To the best of our knowledge, TopicNet is the first work to use a directed graph as the semantic guidance, each node of which is a word, to specify the hierarchical structure of the topics learned from the corpus.
>
> **Concern** on statistical robustness of the reported experimental results.
>
> **Answer:** To demonstrate the performance of our model on the suggested setting, where the number of topics << the number of words << the number of documents, we conduct additional experiment on the RCV1 dataset, which consists of 804,414 documents with a vocabulary size 8000. The number of topics is set as [260, 64, 11, 2]. The results of topic quality are as follows:
>
> |              | Layer-1 | Layer-2 | Layer-3 | Layer-4 |
> |:------------:|:-------:|:-------:|:-------:|:-------:|
> |  GBN SG-MCMC |   0.27  |   0.11  |   0.08  |   0.20  |
> |     WHAI     |   0.12  |   0.18  |   0.12  |   0.18  |
> |    SawETM    |   0.11  |   0.25  |   0.18  |   0.24  |
> | Gauss SawETM |   0.12  |   0.24  |   0.28  |   0.32  |
> |   TopicNet   |   0.20  |   0.24  |   0.29  |   0.36  |
>
> **Q1:** How is the vocabulary curated?
>
> **A1:** We followed common practices to define the vocabulary. For the per-heldout-word perplexity (PPL) experiments, the same as [1], we follow the convention to build a vocabulary by first removing the stop words and then selecting the top K most frequent words, where K=2000, 8000, 10000 for 20newsgroup, RCV1, WIKI, respectively. For document classification and clustering, we remove stop words and then selecting the top K most frequent words, where K=20000, 10000 for 20newsgroup and R8, respectively.
>
> **Q2:** Note that it is a better practice to first learn realistic topic distributions from real text corpora, later sampling various sizes of documents by using the known topic distributions and randomly sampled multinomial topic proportions. Given ground-truth topic distributions and proportions, you will be able to firmly complement the clustering study.
>
> **A2:** Thank you for your suggestion. We are curious whether there exists an appropriate reference that has introduced such an evaluation method. Our concern regarding this evaluation method is that as a latent variable model, the topics and topic proportions are often not identifiable given the observed bag-of-words. Thus, it might not be meaningful to compare the inferred topics and topic proportions with the ground truth ones.
>
> **Q3:** How would the baseline LDA perform if using learned hyperparameters with CGS?
>
> **A3:** We would like to clarify that we have compared with the single-layer DLDA [2], which is equivalent to LDA with learned hyperparameters.
>
> **Q4:** What is exactly the uncertainty and how is it mathematically defined? Do you have any real example where SawETM fails to capture these uncertainties, whereas Gaussian SawETm captures successfully?
>
> **A4:**  Briefly, in SawETM each word and topic is represented as a vector while in Gaussian SawETM each word and topic is represented as a distribution. We refer the semantic uncertainty to the semantic coverage. For example, “fruit” obviously covers more things than "banana", and the Euclidean distance between two vectors can only reflect if "fruit" is related with "banana", while the KL divergence between two distributions can imply which is "fruit" (correspond to the border distribution)
>
> **Q5:** Being explainable and having topic interpretability are not identical in general category.
>
> **A5:** In our work, we have treated “interpretable” and “explainable” in the same way. We will choose “interpretable” to describe our results in the revision.
>
> **Q6:** Topic specificity (how far from the unigram corpus distribution) can often help quality measurement jointly with those two metrics.
>
> **A6:** We appreciate your pointing out the titles of the papers by David Mimno and relevant references for topic specificity. We will study them carefully and add appropriate discussions.
>
> **Q7** WordNet’s hierarchy semantic is mostly limited only to nouns. What happen if some topics are more about actions than entities, thereby being described more by verbs than nouns in their top words?
>
> **A7:** It's true that verbs rarely appear in WordNet's semantic hierarchy. But our model is not limited to nouns, since the topics are learned under the combined influence of the ELBO and regularized loss. Note that the semantic hierarchy only serves as an inductive bias, if it does not provide enough guidance for a topic, then the topic is more likely to be learned from the co-occurrence patterns in the corpus.
>
> [1] Hao Zhang, Bo Chen, Dandan Guo, and Mingyuan Zhou. WHAI: Weibull hybrid autoencoding inference for deep topic modeling. ICLR, 2018:
>
> [2] Yulai Cong, Bo Chen, Hongwei Liu, and Mingyuan Zhou. Deep latent Dirichlet allocation with topic-layer-adaptive stochastic gradient Riemannian MCMC. ICML, 2017.

---

> > ### Comment · Reviewer_s9bs · 2021-08-21
> > **Somewhat clarified but not sufficiently convincing.**
> >
> > I thank the authors for their careful comments.
> >
> > (Concern) The main contributions of the paper are better pinpointed, but theoretical or modeling contributions are not yet strong for NeurIPS submission. The applicational implication must be strong enough.
> >
> > (Statistical Robustness) It is useful to see additional results. However, A vocabulary of 8,000 words is too small to generalize for modern dataset unless it is compared on the level of sub-word embeddings like large-scale language models.
> >
> > (A1) Sometimes vocabulary curation matters the most because if most of the words left in the vocabulary is just a small set of most frequent words, they are likely to co-occur in general texts, and humans can always come up with reasonable interpretations.
> >
> > (A2) Read “A Practical Algorithm for Topic Modeling with Provable Guarantees” by Arora et al. Though this is not a beginning of such realistic evaluation, many papers that this paper cited and that cite this paper adopt semi-synthetic construction. “Prior-aware Composition Inference for Spectral Topic Models” by Lee et al have various resolutions of creating realistic datasets.
> >
> > (A3) It is not readily clear that how DLDA’s performance is comparable to LDA’s with Collapsed Gibbs Sampling with hyperparameter learning option. Do you actually have a result?
> >
> > (A4) While not underestimating the power of distribution representations against the point-vector representation, most papers use Euclidean distance or cosine similarity as a metric to compare two different concepts regardless of their relatedness. Also, it is hard to believe that the proposed distribution representations can actually grasp semantic similarity and uncertainty. It is more likely distributional similarity for which “cat” and “dog” will be more similar than “dog” and “puppy” or “cat” and “kitten”. (Also, do you mean the similarities from A to B and B to A are different by using KL?)
> >
> > (A7) The point is that when you combine ELBO and regularized loss, it will not be easy to flexibly tune the balancing hyperparameter per dataset unless your model could give some guidance based on how much noun hierarchies will be useful for a given corpus. Considering a case of streaming input, it will be even less realistic.

---

> > > ### Author Response · Authors · 2021-08-23
> > > **Additional clarifications**
> > >
> > > We sincerely appreciate your additional comments. We hope our response below could help address your remaining doubts.
> > >
> > >
> > >
> > >  - **A1**: We respectively disagree that a vocabulary size of 8000 is too small to generalize for modern dataset.
> > >
> > >     - According to Zipf’s law, given a large sample of words, the frequency of any word is inversely proportional to its rank in the frequency table, which means the most frequent word will occur about ten times more often as the tenth most frequent word. Thus, the frequency of the terms in the vocabulary will exponentially decay as we move from the most frequent to the least frequent ones.
> > >     - Further increasing the vocabulary size may further improve the performance of downstream tasks, such as classification and clustering, but not necessarily be helpful for enhancing interpretability.
> > >
> > >      - The table below shows a set of words (after removing stopwords) and their ranks and normalized frequencies.
> > >
> > > | Ranks          | 1| 500           |1000          |2000 |3000 |4000 | 5000|6000| 7000| 8000|
> > > | ------------- |:-------------:|:-------------:| :-------------:|:-------------:|:-------------:|:-------------:|:-------------:|:-------------:|:-------------:|:-------------:|
> > > | **Words**      | percent| Aim	|determin|	houston|	pa|	anxi|	flur|	encrypt|	citr|	billy
> > > | **Count**      | 884475| 38574|	18941|	7213|	3801	|2298|	1563	|1140|	868	|693
> > > |  **Normalized frequencies**     |  | 1.0 |0.49| 0.18 | 0.09| 0.05| 0.04| 0.03 |0.02 | 0.02
> > >
> > >
> > >  - **A2**: We appreciate your valuable advice. We have read these two recommended papers and now understood what you meant by using semi-synthetic documents for evaluation.
> > >
> > >     - The basic pipeline is to fix the topics $\Phi$, sample random topic proportions and document lengths, and combine them to generate synthetic bag-of-words documents. Then a topic model of choice is run on these semi-synthetic documents to extract the underlying topics, which are then mapped to the ground truth topics via some techniques, such as solving linear programs, to computing a matching error.
> > >
> > >      - There are three important reasons we have not yet followed this type of evaluation approach:
> > >          -	There is a non-identifiability issue in topic models (or any latent-variable models), making us question whether it is that meaningful to compare the extracted latent factors to the ground truth. For example, if using Gibbs sampling, each MCMC step will produce a different estimate of $\Phi$, but it is theoretically problematic to average over these $\Phi$’s collected at different MCMC steps.
> > >          -	Even if the comparison between the extracted and “ground-truth” topics is meaningful, it is not simple to define an easy-to-compute metric.
> > >         -	Our models build a topic hierarchy, making it even more difficult to apply this evaluation method developed for a shallow topic model.
> > >
> > >
> > >  - **A3**: For a single-layer DLDA, we did use collapsed Gibbs (See Eq. (28) in [1], where $\phi_{k:}^{(2)}\theta_j^{2}$ is replaced with $r_k$). This collapsed Gibbs sampling step is exactly the same as LDA with collapsed Gibbs except that we impose a gamma prior on each $r_k$ and sample its posterior from gamma (after using latent counts based variable augmentation). By contrast, in LDA, one has a single concentration parameter $\alpha$ for all $K$ dimensions that will be updated. The role of $\alpha$ in LDA is the same as $\sum_k r_k$ in the single-layer DLDA. LDA is using a symmetric Dirichlet prior while the single-layer DLDA is using an asymmetric Dirichlet prior (one can write $r_k$ as $\alpha* r_k/\alpha$, where $\alpha$ has a gamma prior and $(r_1,…,r_K)/\alpha$ has a Dirichlet prior.
> > >
> > >
> > >
> > >
> > >  - **A4**: Distribution representations can be exemplified as a soft region, where more specific words (e.g., puppy and kitten) have smaller uncertainties, while those denoting broader concepts (e.g., cat and dog) have larger uncertainties [2]. Furthermore, distribution representations allow the entailment relationship to be expressed naturally, with general words such as “cat” corresponding to broad distributions that encompass more specific words such as “kitten”. Specifically, KL divergence is a natural energy function for representing entailment between concepts: a low KL divergence from x to y indicates that we can encode y easily as x, implying that y entails x. Through encapsulation of probability densities, it can intuitively reflect the hierarchical structure of semantic concepts, thus offering us an effective way to model the topic hierarchy.
> > >
> > >
> > >  - **A7**:  We'd like to clarify we have simply fixed this balancing hyperparameter as one in this paper.
> > >
> > >      - If there is a specific downstream task, then it is simple to tune this single balancing hyperparameter per dataset via cross-validation.
> > >
> > >      - As discussed in "Clarification on the main objective of injecting the semantic graph prior," the noun hierarchies provide inductive bias to guide the learning, and whether it is useful for a given corpus will be depending on whether this injected inductive bias aligns with the final objective.
> > >
> > >      - If it hurts the specified downstream task, then a simple solution would be setting the balancing hyperparameter to zero, which is likely to be achievable by cross-validation.
> > >
> > > [1] Zhou M, Cong Y, Chen B. Augmentable gamma belief networks. JMLR 2016.
> > > [2] Vilnis L, McCallum A. Word representations via gaussian embedding. ICLR 2015.

---

### Review · Ethics_Reviewer_9a4y · 2021-08-12

**Recommendation:** see above

**Ethics Review:**

I believe this was flagged only for an example which the authors have committed to replace.

---

### Review · Ethics_Reviewer_2KXV · 2021-08-13

**Recommendation:**

I see no further issues with this work, and believe the current ones can be easily addressed.

**Ethics Review:**

In reading this paper, it seems that the key point of contention is the example in Fig 1 where there seems to be an implicit categorization of a particular set of topics. Overall, the figure shows an illustrative on a basis of implicit associations; however, the perceived goal here is not malicious or intentional.

---

### Author Response · Authors · 2021-08-10
**Clarification on the main objective of injecting the semantic graph prior**

We would like to thank all four reviewers for providing insightful and very detailed comments and suggestions, which will be very helpful for us to revise our manuscript.

A shared concern among the reviewers is that injecting a pre-defined semantic graph to guide the topic discovery may sometimes lead to worse performance in certain downstream tasks, such as document clustering and classification. Another concern is that doing so may impose concepts not sufficiently supported by the corpus.

To address both concerns, we'd like to clarify that if a pre-defined semantic graph is injected to guide the discovery of hierarchical topics, then the main objective is often to reconstruct each concept in the semantic graph as a distribution of words (i.e., a topic) given the training corpus.

- Imposing this inductive bias could help correct data biases, help contextualize the meaning of a concept in the training corpus, and help discover a specific topic hierarchy that would otherwise be shadowed by the dominant topics of the training corpus.
  - A potential data debiasing example: Suppose the training corpus has a biased view of "family" as the composition of "mother," "father," and "son," then we can build a semantic graph, which includes "family" as a parent node and "mother," "father," "son," and "daughter" as its child nodes. Injecting this semantic graph to guide the topic discovery could help correct the data bias of underrepresenting "daughter" as an essential component of the “family'' concept.
- When the semantic graph is not well matched to the main themes of the training corpus, injecting it as a prior will undoublty hurt some downstream tasks, such as document classification and clustering. However, this does not necessarily mean worse performance if our goal is not to boost the performance on these downstream tasks, but to better discover the concepts present in the semantic graph. For example, injecting a semantic graph on "religion'' and contextualizing it under 20Newsgroup will probably lead to bad performance in terms of categorizing the documents into 20 different categories, but may lead to better topic representations of various concepts of religion in 20newsgroup.

---

### Decision · Program_Chairs · 2021-09-27

**Decision:**

Accept (Poster)

**Comment:**

This paper develops TopicNet, a generative model that can learn hierarchical topics by incorporating a pre-defined semantic graph. The main contribution of the paper seems to be on the application side, and all reviewers agreed the paper is above the acceptance threshold.

I would like to encourage the authors to incorporate the reviewers' feedback in their revised version. In particular:
+ Incorporate the new results from the response to reviewer aZKr.
+ Add the metrics of inter-topic (dis)similarity and topic specificity.
+ Add an experiment to showcase that the method can benefit from prior knowledge in terms of both interpretability and performance on downstream tasks.
+ Add a more detailed figure to illustrate the Gaussian SawETM and TopicNet models.
+ Clarify the other points raised in the reviews.